# SEMANTIC PARALLELISM: REDEFINING EFFICIENT MOE INFERENCE VIA MODEL-DATA CO-SCHEDULING

**Yan Li**[*]
Huawei Technologies
liyan412@huawei.com

**Zhenyu Zhang**[*]
Sun Yat-Sen University
zhangzhy239@mail2.sysu.edu.cn

**Zhengang Wang**
Huawei Technologies
wangzhengang@huawei.com

**Pengfei Chen**
Sun Yat-Sen University
chenpf7@mail.sysu.edu.cn

**Pengfei Zheng**[†]
Huawei Technologies
zhengpengfei18@huawei.com

## ABSTRACT

Prevailing LLM (Large Language Model) serving engines employ expert parallelism (EP) to implement multi-device inference of massive Mixture-of-Experts (MoE) models. However, the efficiency of expert parallel inference is largely bounded by inter-device communication, as EP embraces expensive all-to-all collectives to route tokens to the remote experts if not collocating on the same GPU/NPU device. Nevertheless, state-of-the-art schemes treat expert device-placement and request (or token) device-scheduling as separate concerns, triggering excessive communication between them and compromising inference efficiency

This paper proposes *Semantic Parallelism*, a novel parallelism paradigm that minimizes the steep communication costs in EP-centric MoE serving via model-data collaborative scheduling. We implement *Semantic Parallelism* in a framework called Sem-MoE. Sem-MoE maximally collocates experts and their activating tokens onto the same device using proactively modeled activation likelihood between them and introduces three key techniques: (1) Offline model scheduling, which preliminarily clusters and collocates experts onto devices based on their co-activation tendencies for certain classes of input. (2) Online inter-request data scheduling for Attention-DP setups, which proactively rebatches incoming requests onto the device that hosts experts most likely and frequently activated by the corresponding requests. (3) Online intra-request data scheduling for Attention-TP setups, which seamlessly fuses a token reshuffling procedure into the original inference pipeline and proactively reschedules tokens to devices to reduce dispersed remote routing. We build Sem-MoE into a prevailing LLM serving engine SGLANG. Experiments show our collaborative scheduling approach can effectively reduce the all-to-all communication volume in EP and achieve superior inference throughput compared to existing solutions.

## 1 INTRODUCTION

The democratization of large language models (LLMs) has been largely driven by continuous model scaling. Over the past five years, the parameter count of the largest trained LLMs has increased by three orders of magnitude, posing significant challenges to the scalability and economic viability of both training and inference under modern AI hardware constraints.

---

[*]Equal contribution
[†]Corresponding author

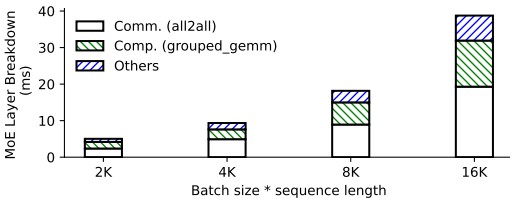

Figure 1: Latency breakdown for DeepSeek-V2-Lite inference over a single MoE layer. Hardware: 8-GPU server with fast inter-GPU network (specialized connection with over 400GB/s bandwidth).

To mitigate these challenges, the Mixture-of-Experts (MoE) architecture Fedus et al. (2022); Artetxe et al. (2022); Jiang et al. (2024) has been introduced. Unlike dense models, MoE models sparsely activate one or more expert sub-networks per input, enabling training of trillion-parameter models without compromising accuracy, while maintaining a sub-linear increase in computational cost. This approach has gained widespread adoption in recent industrial-strength LLMs, including DeepSeek-V3 DeepSeek-AI (2024b)/DeepSeek-R1 DeepSeek-AI (2025), GPT-OSS OpenAI et al. (2025), the Qwen3-Series Yang et al. (2025), and Kimi-K2 Team et al. (2025).

However, at inference time, massive MoE models still require substantial GPU/NPU[1] resources to compute, store, and load both expert and attention parameters. To achieve scalability and meet latency requirements, existing inference frameworks deploy multi-dimensional parallelism strategies that distribute experts and attention blocks across interconnected devices. An efficient parallelization scheme must effectively partition input tokens and model parameters, maximize resource utilization, and minimize communication overhead.

To address the memory demands of large-scale MoE deployment and leverage aggregate memory bandwidth, modern inference engines such as SGLang Zheng et al. (2024) and vLLM Kwon et al. (2023) employ expert parallelism (EP), whereby experts are distributed across devices. Attention layers are typically parallelized via data parallelism (DP) or tensor parallelism (TP). While EP enables parallel computation of experts across GPUs, it introduces significant communication overhead: intermediate activations must be *dispatched* from the gating module on a source GPU to the destination GPUs hosting the routed experts, and later *combined* back after expert computation. These operations often result in cluster-wide any-to-any token shuffling, typically implemented via two `all2all` collective operations (e.g., NCCL/HCCL's `all2all`).

Our analysis reveals that the inference performance of MoE models remains severely constrained by these costly `all2all` operations. For instance, a preliminary experiment running SGLang on the DeepSeek-V2-Lite model with 8 GPUs shows that EP communication accounts for up to 59.2% of the forward-pass latency in the MoE layers, respectively—even on high-speed interconnects (see Figure 1). This bottleneck is further exacerbated on slower interconnects such as PCI-e or Ethernet. Therefore, systematically reducing EP communication has become a critical task for improving the efficiency and scalability of MoE inference.

In this paper, we demonstrate that the communication overhead of EP can be substantially reduced through a novel **semantic-aware model–data collaborative scheduling** approach, namely *Semantic Parallelism*. This method forecasts expert routing paths for both requests and individual tokens, and proactively co-schedules tokens and experts to eliminate redundant communication. We implement the above idea in a system **Sem-MoE**, including two key techniques:

First, Sem-MoE performs *offline model scheduling* to reduce expert dispersion. Experts that are frequently activated together are clustered and placed on the same device or server based on predicted token-expert affinities. This grouping is performed periodically offline to avoid runtime overhead.

Second, Sem-MoE employs *online data scheduling* to align input tokens with their corresponding expert groups. This includes: (1) *Inter-request scheduling* for DP-based attention: dynamically batching requests to maximize expert affinity and minimize cross-device transfers. (2) *Intra-request scheduling* for TP-based attention: proactively shuffling token activations during the TP communication phase. Specifically, Sem-MoE replaces the standard post-attention `allreduce` with a

---

[1]We use GPU and NPU interchangeably in this paper.

`shuffled-reduce-scatter` and a deferred `shuffled-allgather`, effectively merging proactive token routing with necessary data transformation.

Through collaborative model-data scheduling, semantic parallelism significantly reduces communication volume and improves inference throughput, as demonstrated through extensive experiments implemented on top of SGLang.

We list the contributions of this paper as follows.

1. We conduct a comprehensive data analysis and reveal a significant *context-independent correlation* between tokens and experts in large-scale MoE models, which provides a foundational insight for optimizing expert placement and token routing.

2. We design and implement an efficient *model-data collaborative scheduling algorithm* that leverages the observed token–expert affinity. Our scheduler improves local activation rate by **15.4%** compared to baseline methods, substantially reducing unnecessary cross-device communication.

3. We implement semantic parallelism in **Sem-MoE** on top of the state-of-the-art inference engine SGLang and perform extensive evaluations. The results demonstrate that Sem-MoE achieves a throughput improvement of up to **2.78x** under specific SLOs in Attention-DP scenarios and up to **24.9%** latency reduction under Attention-TP setups, validating the practical effectiveness of our approach.

## 2 BACKGROUND

**Mixture-of-Experts** The Mixture-of-Experts (MoE) architecture is a conditional computation paradigm designed to scale model capacity without a proportional increase in computational cost [1]. Unlike dense models, where all parameters are activated for every input, an MoE model consists of a multitude of expert sub-networks (typically Feed-Forward Networks, FFNs) and a gating network (or router). For each input token, the gating network predicts a sparse combination of experts (e.g., the top-$k$ experts) to which the token is dispatched. Only the selected experts are activated for computation. The most common gating function is the Top-K Gating, which selects the $k$ experts with the highest scores. This design enables models to possess a vast number of parameters (e.g., trillions) while keeping the FLOPs per token roughly constant, as only a small, fixed number of experts (e.g., $k = 2$) are active per token. This has made MoE the de facto standard for building state-of-the-art large language models, such as the DeepSeek series DeepSeek-AI (2024a;b; 2025), the GPT-OSS series OpenAI et al. (2025), and the Qwen series Qwen-Team (2024); Yang et al. (2025).

**MoE Training Systems.** There has been extensive research on optimizing systems of MoE training systems, including FastMoE He et al. (2021), FasterMoE He et al. (2022), TA-MoE Chen et al. (2022), SmartMoE Zhai et al. (2023), and FlexMoE Nie et al. (2023). However such optimizations can not directly translate to inference scenarios as inference is workload-sensitive and strongly emphasizes latency over throughput.

**MoE Inference Systems.** Integrated serving engines such as DeepSpeed-MII Holmes et al. (2024), TensorRT-LLM NVIDIA, vLLM Kwon et al. (2023), and SGLang Zheng et al. (2024) have holistic optimization for LLM inference that spans serving schedulers (e.g., continuous batching), dedicated high-performance kernels, efficient parallelization, quantization, and elaborate compiler passes for graph-level optimizations. Built upon these general holistic optimizations for LLM inference, DeepSpeed-MoE Rajbhandari et al. (2022); Singh et al. (2023) and Tutel Hwang et al. (2022) specifically optimize MoE models' computation and communication. Following the design paradigm of DeepSpeed-MoE, popular industry and open-sourced inference engines like vLLM and SGLang also adopted expert parallelism deployment. Sem-MoE specializes in optimizing MoE parallelization (particularly EP) and inherits holistic optimizations from prior work.

**MoE Load-balancing and Experts Re-grouping.** Lina Li et al. (2023) probes the variation of expert hotness and allots non-uniform expert replicas to achieve load-balanced expert computation. Similar studies Huang et al. (2023) exist to pursue expert load balancing and mitigate other sources of MoE computing inefficiencies. EPS-MoE Qian et al. (2025) optimizes the computation of MoE FeedForward Network (FFN) modules by dynamically selecting the best backend implementation of GroupGemm and DenseGemm. DeepSeek also adopts EPLB (expert-parallelism load balancing)

in its real-world deployment DeepSeek-AI (2024b). ExFlow Yao et al. (2024) exploits the affinity between experts across adjacent layers to reduce remote routing and collocate closely related experts. Exflow only considers the model scheduling for MoE models, and requires a heavy `allgather` before the execution of each model layer, which significantly incurs memory pressure and extra communication overhead. MoETuner Go & Mahajan (2025) optimizes the MoE model serving by finding an optimal expert placement strategy to minimize inter-device communication.

**MoE Offloading and Prefetching.** Existing prediction-based work on MoE inference primarily focuses on prefetching offloaded experts and strategically saving GPU memories Yi et al. (2023); Xue et al. (2024); Zhong et al. (2024), though offloading can extend inference latency and is rarely used in latency-critical serving scenarios. In contrast, Sem-MoE focuses on the speculative reduction of communication overheads and exposes no risks to compromise latency. The work also constructs probabilistic models to predict the token-expert routing paths, while Sem-MoE's features modelling more comprehensive MoE information, i.e., intra-layer and inter-layer expert affinity and token-expert affinity, compared to prior work. Pre-gated MoE Hwang et al. (2024) modifies the MoE model architecture to predict the experts to route at the next layer. Sem-MoE requires no modification to MoE architecture.

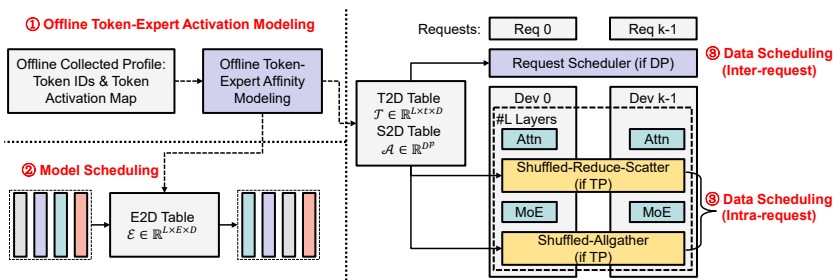

Figure 2: The workflow of semantic parallelism in Sem-MoE.

# 3 METHODOLOGY

## 3.1 SEMANTIC PARALLELISM: OVERVIEW

Figure 2 illustrates the overall workflow of *semantic parallelism* in Sem-MoE. Solid and dashed lines in the figure represent online and offline operations, respectively. The process begins with collecting token activation profiles, which include token identifiers and token-expert activation frequencies (Step ① in Figure 2). Based on these profiles, Sem-MoE probabilistically model the token-expert routing likelihood and formulates a balanced token-expert co-clustering problem to generate scheduling hints. These hints are materialized as lightweight lookup tables: a token-to-expert-group table $\mathcal{T}^2$, an expert-group-sequence-to-expert-group table $\mathcal{A}$, and an expert grouping table $\mathcal{E}$.

These scheduling tables drive the subsequent collaborative model-data scheduling. In the model scheduling phase (Step ②), Sem-MoE utilizes the expert-to-device table $\mathcal{E}$ to reconfigure the placement of experts across all layers. In the data scheduling phase (Step ③), different policies are applied depending on the parallelism strategy of the attention layers:

- For attention layers deployed with Data Parallelism (DP), Sem-MoE employs *inter-request* data scheduling. This policy reorders incoming requests according to the token-to-device table $\mathcal{T}$ to maximize request-expert-group affinity, thereby reducing the `all2all` communication overhead across DP domains.

- For attention layers partitioned via Tensor Parallelism (TP), Sem-MoE adopts *intra-request* data scheduling. This technique proactively shuffles tokens during the post-attention `reduce-scatter` operation, directing them to devices predicted to host their target experts in advance, thus minimizing potential token redistribution in subsequent MoE layers.

---

[2]We use expert group and expert cluster interchangeably.

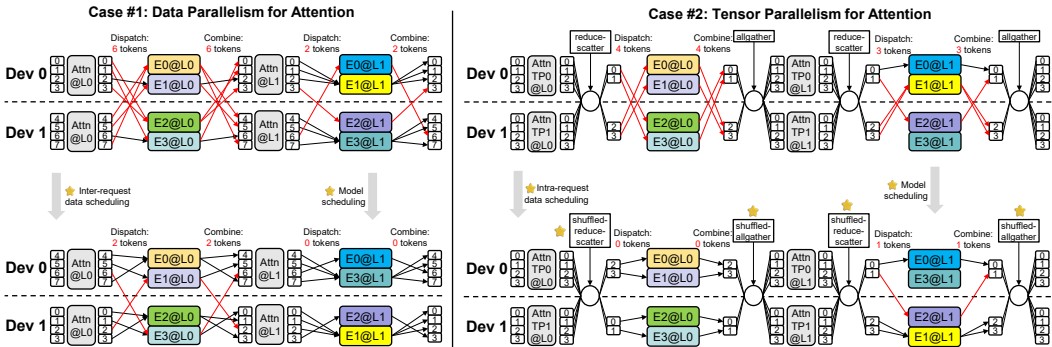

Figure 3: An illustrating example of Sem-MoE. In Case #1 (Attention with DP), Sem-MoE reschedules requests and adjusts expert placement in Layer 1, reducing the number of remotely activated tokens (Remote activated tokens refer to the total number of tokens dispatched to and combined from remote devices) from 16 to 4. In Case #2 (Attention with TP), token rescheduling via `shuffled-reduce-scatter` and expert repositioning in Layer 1 reduce remote token activations from 12 to 2.

Figure 3 provides a concrete example of Sem-MoE's operation. In the baseline of Case 1 (top row), requests are distributed across DP ranks for independent attention computation. After expert assignment, tokens are dispatched to their respective expert devices via `all2all` operations. With Sem-MoE's inter-request scheduling, requests are intelligently mapped to DP ranks to enhance data locality, reducing `all2all` volume. This effect is further amplified by complementary model scheduling that optimizes expert placement. In Case 2, which involves TP for attention, tokens require reduction before dispatch and gathering after combination. Sem-MoE's intra-request scheduling predicts expert routes prior to the gating module, allowing tokens to be shuffled and scattered via a customized `shuffled-reduce-scatter (SRS)` operator. Combined with model scheduling, this approach achieves a higher local activation rate, significantly cutting down `all2all` traffic.

The inference acceleration achieved by Sem-MoE stems from the increased local activation rate enabled by collaborative model-data scheduling. Let $G$ denote the number of devices, $B$ the global batch size, $S$ the sequence length, and $k$ the number of experts activated per token. The communication volume of an `all2all` operation is given by $\frac{\alpha k B S}{G}$, where $\alpha$ represents the fraction of non-local activations. By maximizing the local activation rate (i.e., minimizing $\alpha$), Sem-MoE effectively trims communication overhead. The subsequent sections detail the offline modeling and online scheduling algorithms.

## 3.2 PREDICTING EXPERT ROUTING PATH

The routing choice of each MoE layer is given by the gating function: $G_L(h_{L,j}) = \texttt{top-k}(\texttt{softmax}(\mathbf{W}_{L,g}h_{L,j} + \mathbf{b}_{L,g}))$. Accurate prediction of token-expert routing patterns in advance is fundamental to Sem-MoE's scheduling optimization.

**Context-Independent Token Activation Prediction.** We observe that despite the theoretical dependence of expert routing on contextual semantics (as expressed by the gating function $G_L(h_{L,j})$), in practice, tokens exhibit strong *context-independent* affinities to specific experts. Table 1 presents three metrics illustrating the activation concentration within DeepSeek-V2-Lite. By profiling the model on the ShareGPT Datasets (2023) dataset, we record the intermediate top-k activation statistics. We then calculate the F1-score for predicting the activated experts in each layer by relying solely on the most frequently activated (hottest) top-k experts. Additionally, we compute the cumulative hotness of these top-k experts alongside the maximum hotness observed among the remaining non-top-k experts. For all evaluated metrics, the P25, P50, and P75 quantiles are reported.

For any specific token, the distribution of expert activation across diverse contexts is remarkably skewed yet stable: a given token consistently routes to a narrow, static subset of *top-k experts*, leaving the majority of experts largely dormant. Throughout all layers of DeepSeek-V2-Lite, the median (p50) cumulative hotness of the top-k experts ranges from 0.833 to 0.976. In contrast, the maximum

Table 1: Empirical study of token activate patterns based on DeepSeek-V2-Lite.

| Layer ID | F1-score | | | Cum. Hotness (Top-k) | | | Max. Hotness (Non-top-k) | | |
|---|---|---|---|---|---|---|---|---|---|
| | p25 | p50 | p75 | p25 | p50 | p75 | p25 | p50 | p75 |
| 1 | 1.000 | 1.000 | 1.000 | 0.833 | 1.000 | 1.000 | 0.000 | 0.042 | 0.071 |
| 6 | 0.833 | 1.000 | 1.000 | 0.767 | 0.917 | 1.000 | 0.000 | 0.056 | 0.083 |
| 11 | 0.833 | 1.000 | 1.000 | 0.722 | 0.889 | 1.000 | 0.000 | 0.056 | 0.083 |
| 16 | 0.667 | 0.833 | 1.000 | 0.708 | 0.889 | 1.000 | 0.000 | 0.053 | 0.083 |
| 21 | 0.667 | 0.833 | 1.000 | 0.667 | 0.833 | 1.000 | 0.000 | 0.050 | 0.074 |
| 26 | 0.667 | 0.833 | 1.000 | 0.667 | 0.833 | 1.000 | 0.000 | 0.051 | 0.083 |

hotness recorded among the cold experts remains negligible—hovering near 0.05 at p50—which highlights a robust and pronounced disparity in activation. Finally, § A.2 in the Appendix expands upon this phenomenon across more scenarios, confirming its broad generalizability.

The above phenomenon enables effective prediction based solely on token identity. Through offline profiling on datasets such as *Sharegpt* using models including DeepSeek-V2-lite and Qwen3-30B-A3B, we construct a **token-to-expert activation table** $\mathbf{T}^{(L)} \in \mathbb{N}^{t \times N^{(L)}}$ for each MoE layer $L$, where $\mathbf{T}_{j,k}^{(L)}$ counts how frequently token $x_j$ activates expert $E_k^{(L)}$. The corresponding routing probability is: $\Pr(E_k^{(L)}|x_j) = \mathbf{T}_{j,k}^{(L)} / \sum_{k=1}^{N^{(L)}} \mathbf{T}_{j,k}^{(L)}$. For efficient online inference, these probabilities are tabulated in a **token-to-expert confidence table** $\mathcal{C}_p \in \mathbb{R}^{t \times N}$. Out-of-vocabulary tokens are handled via nearest-neighbor matching in the embedding space.

The token-level predictions form the basis for scheduling in both Attention-DP and Attention-TP scenarios. For **Attention-DP**, the affinity of an entire request to an expert group is derived by aggregating the predictions of its constituent tokens, enabling request-level scheduling. For **Attention-TP**, the fine-grained token-level predictions are directly utilized, and are further refined by modeling inter-layer dependencies, as discussed in Section 3.3. This predictive framework provides the essential guidance for Sem-MoE's collaborative scheduling optimization detailed next.

### 3.3 MODEL-DATA COLLABORATIVE SCHEDULING

We illustrate how such expert-routing forecasting models can guide the co-dispatching of tokens and experts. Sem-MoE formulates the model-data co-scheduling problem as a 0-1 integer programming (ILP) -based co-clustering problem.

From the offline profiling, we obtain the number of deduplicated (un-deduplicated) tokens $t$ ($S$), the number of experts per layer $N$, the number of clusters $E$ (also EP degree), the token $j$ frequency $\boldsymbol{a}_j$, and the activation probability $\mathcal{C}_{p,jk}$ that token $j$ activates expert $k$. The decision integer variables are set as the routing $\boldsymbol{R}_{ij} \in \{0, 1\}$ of token $j$ to cluster $i$, and the placement $\boldsymbol{C}_{ij} \in \{0, 1\}$ of expert $j$ to cluster $i$.

We aim to minimize an objective function $\mathcal{L} = \theta \sum_{i=1}^{E} \left| \sum_{j=1}^{t} (\boldsymbol{R}_{ij}\boldsymbol{a}_j) - \frac{S}{E} \right| + (1 - \theta) \sum_{i_1 \neq i_2} \left( \sum_{j=1}^{t} \sum_{k=1}^{N} (\boldsymbol{R}_{i_1 j}\boldsymbol{C}_{i_2 k}\mathcal{C}_{p,jk}\boldsymbol{a}_j) \right)$, where the left part is to ensure that the token frequencies of different clusters as even as possible to promote load balancing among EP ranks, and the right part is to minimize the all2all communication overhead caused by remote activation (i.e., the summation of all the activations of tokens and experts belonging to different clusters), a factor $\theta \in (0, 1)$ controlling the percentage of two sub-objectives. We further require that each token belongs to only one class, each expert belongs to only one class, and the number of experts in each class is equal by adding hard constraints $\sum_{i=1}^{E} \boldsymbol{R}_{ij} = 1$, for $j = 1 \ldots t$, $\sum_{i=1}^{E} \boldsymbol{C}_{ij} = 1$, for $j = 1 \ldots N$, and $\sum_{j=1}^{N} \boldsymbol{C}_{ij} = \frac{N}{E}$, $i = 1 \ldots t$.

The above ILP problem is difficult to solve directly using LP solvers, given a large number of intermediate variables introduced in the linearization process. Sem-MoE provides an alternating optimization algorithm. It can quickly obtain a feasible solution while ensuring load balancing. The detailed co-clustering algorithm can be referred to in § B in the Appendix. The solution can then be applied to offline model scheduling and online inter-/intra-request data scheduling.

**Model scheduling.** Before deployment, Sem-MoE adjusts the expert placement layout according to the solved $C$, placing expert $j$ to device $k$ if $C_{jk} = 1$. Accordingly, Sem-MoE shuffles the column of the gate matrix, thereby realizing a transparent expert re-distribution.

**Data scheduling: Attention-DP Scenarios.** In Attention-DP setups, where requests are processed independently across DP ranks, scheduling operates at the *request granularity*. We use the variable $S_r \in \llbracket E \rrbracket$ to denote the cluster to which request $r$ needs to be scheduled. Once the scheduling of tokens is determined, the scheduling of the request $r$ can be determined by aggregating the routing result of its tokens, $S_r = \arg\max_{j \in \llbracket E \rrbracket} \sum_{i \in r} R_{ij}$. Meanwhile, to achieve runtime load balance, Sem-MoE realizes a workload-aware balanced request scheduling algorithm. For continual $E$ requests, Sem-MoE guarantees these requests are distributed to all $E$ ranks, such that the loads of all ranks would not skew in decoding stage. Detailed algorithm could be referred to in Algorithm 2 in § B. This request-level scheduling minimizes cross-device communication by collocating entire requests with their most likely expert group (i.e., DP rank).

**Data Scheduling: Attention-TP Scenarios.** In Attention-TP setups, the attention computation itself is distributed, requiring fine-grained, *token-level* scheduling. Here, we enhance the basic token-expert prediction with **inter-layer expert-expert affinity**. We observe that expert selections exhibit Markovian dependencies across layers: the experts chosen at layer $L$ depend on selections at previous layers. We model this using an $n$-gram device transition model: $\Pr(D_k^{(L)} | D^{(L-1)}, \ldots, D^{(L-n)})$ where $D^{(l)} \in \{1, \ldots, Q\}$, where $D^{(l)} \in 1, \ldots, Q$ denotes the device index of the expert selected at layer $l$. These transitions are stored in an **expert-group-sequence-to-expert-group confidence table** $\mathcal{A}_p$ (we use 2-gram in practice). Together with the token routing matrix $R$, we can achieve more accurate proactive scheduling during the TP communication phase. The detailed algorithm can be found in Algorithm 3.

### 3.4 IMPLEMENTATION AND SYSTEM OPTIMIZATION

Sem-MoE is implemented as a plug-in module for the SOTA LLM inference engine SGLang. Our system comprises approximately 5,000 lines of Python code, along with several custom Triton OpenAI kernels for high-performance communication operations.

To support affinity-aware scheduling in the Attention-DP scenario, we extend SGLang's request scheduler to incorporate token–expert affinity information derived from our prediction models. This enables the runtime to batch requests with similar expert activation patterns onto the same device, minimizing cross-device communication.

For the Attention-TP scenario, we implement two fused communication primitives: `shuffled-reduce-scatter (SRS)`, and `shuffled-allgather (SAG)`. These kernels integrate speculative token shuffling—based on predicted expert routes—into standard `reduce-scatter` and `allgather` collectives. The shuffling logic relies on an optimized `argsort` kernel, which outperforms the native PyTorch implementation by **25%**. The overall overhead of embedding shuffling into the ring-based communication schedule is negligible, measured at approximately **1%**. Furthermore, for efficient `all2all`, Sem-MoE integrates frontier MoE communication libraries DeepEP Zhao et al. (2025).

By combining offline expert reorganization with online token- and request-level scheduling, Sem-MoE achieves significant reductions in `all2all` communication volume, leading to improved end-to-end inference throughput in both DP and TP configurations.

## 4 EXPERIMENT

### 4.1 EXPERIMENTAL ENVIRONMENTS

We evaluate Sem-MoE on an 8-GPU server, representing commercial GPU servers unified for both training and inference, which are configured with 96GB-HBM per GPU and fast homogeneous interconnects. GPUs inside a server can communicate with each other at a premium bandwidth (specialized network with over 400GB/s bandwidth). The server is equiped with two 44-core Intel CPUs and 2TB DDR5 memory.

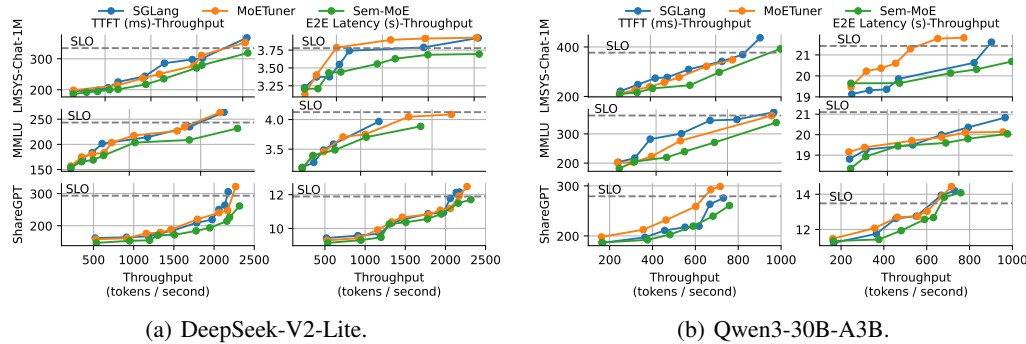

(a) DeepSeek-V2-Lite.  (b) Qwen3-30B-A3B.

Figure 4: Attention-DP Scenario: Inference throughput under TTFT and E2E latency SLOs.

## 4.2 MODELS, DATASETS AND WORKLOAD TRACES

**Models.** We choose two types of typical MoE models for evaluation, i.e., Qwen3-30B-A3B with 128 experts per layer and DeepSeek-V2-Lite with 64 routed experts per layer. Both Qwen3-30B-A3B and DeepSeek-V2-Lite are prevailing open-sourced MoE models.

**Datasets.** We use the following three representative datasets *MMLU* Hendrycks et al. (2021b;a), *lmsys-chat-1m* Zheng et al. (2023), *ShareGPT-Vicuna-unfiltered* Datasets (2023). In the experiments, we only focus on the prompt parts of these datasets. These datasets contain data from different domains, representing real-world user request patterns, and can effectively evaluate the affinity of requests and tokens from different domains for experts.

## 4.3 BASELINES AND PERFORMANCE METRICS

We select SGLang and MoETuner as the baselines to compare, representing the SOTA LLM inference engine and SOTA MoE model scheduling technique.

**SGLang:** SGLang Zheng et al. (2024) is a prevailing open-source LLM inference framework, incorporating numerous optimizations, including but not limited to continuous batching, paged attention, flash attention, radix-attention, advanced quantization, etc. SGLang declares optimizations for MoE models with high-performance, fused triton kernels OpenAI, supporting DP and TP for attention layers and EP for MoE layers. SGLang is the SOTA open-sourced LLM inference engine, which we set as a strong baseline to compare.

**MoETuner:** MoETuner Go & Mahajan (2025) is an optimization framework that enhances MoE model serving performance by finding an optimal expert placement strategy. It addresses critical bottlenecks in expert parallelism, namely imbalanced token processing loads across GPUs and skewed inter-GPU communication, which lead to significant tail latency. The core of MoETuner is an Integer Linear Programming (ILP) formulation that leverages predictable token routing dependencies across layers. It jointly optimizes expert-to-GPU assignments to balance computational workloads and minimize communication costs, thereby reducing end-to-end execution time. We embed MoETuner into SGLang as a comparable baseline.

**Metrics:** To mitigate performance fluctuating, we set the input length and output length fixed. Then we vary the request rate from 10 to 175 req/s, and observe the following metrics. **Throughput** is the number of tokens (tokens/s) that an inference system can process per unit time. **TTFT (Time to First Token)** measures the duration between the request's arrival and the first token's generation time. **E2E (End-to-End) Latency** measures the duration between the request's arrival and the last token's generation time. Following prior work Wu et al. (2024b;a), we set the latency SLO as 5× of the latency under the lightest input load (minimal request rate). For each dataset, we use 20% of the data to train the token activation prediction model and generate the expert placement table, and the remaining 80% is used to sample experimental requests.

| Models | Input Length | p99 TTFT (ms) | | | Median E2E Latency (ms) | | |
|--------|--------------|-------|----------|---------|--------|----------|---------|
| | | SGLang | MoETuner | Sem-MoE | SGLang | MoETuner | Sem-MoE |
| DeepSeek-V2-Lite | 256 | 84.87 | 80.96 | **75.63** | 617.87 | 604.46 | **603.10** |
| | 512 | 98.10 | 92.17 | **88.69** | 609.81 | 602.02 | **599.89** |
| | 1024 | 111.19 | 119.76 | **100.74** | 608.96 | 606.54 | **604.45** |
| Qwen3-30B-A3B | 256 | 85.72 | 87.24 | **74.46** | 758.60 | 763.97 | **750.26** |
| | 512 | 104.76 | 101.86 | **83.87** | 766.80 | 759.41 | **759.16** |
| | 1024 | 107.73 | 107.36 | **103.69** | 769.83 | 764.16 | **762.30** |

Table 2: Attention-TP Scenario: TTFT and E2E latency under different input lengths.

## 4.4 END-TO-END INFERENCE PERFORMANCE

Figure 4 shows the end-to-end performance evaluation of Sem-MoE and two baselines.

**Attention-DP Scenario.** For the attention-DP scenario, requests are scheduled to different DP ranks for attention and synchronized at MoE layer. The latency of each layer is determined by the slowest DP(EP) rank. Thus we use token throughput to measure the overall performance in attention-DP. We draw a latency-throughput curve to measure the highest throughput a system can achieve under pre-defined SLOs, shown in Figure 4. Data points near the bottom-right corner are better. For Deepseek-V2-Lite, Sem-MoE achieves throughput improvements of 31% and 221% against SGLang with DeepEP under TTFT and end-to-end latency SLO constraints, and 32% and 278% against MoETuner, respectively. For Qwen3-30B-A3B, Sem-MoE's throughput improvement peaks at 98% and 11% against SGLang under SLO constraints, while also achieving gains of 35% and 32% against MoETuner. As the request rate increases continually, baselines stock a bunch of unprocessed requests, yielding a steeper curve, resulting in the above high throughput improvement (221% and 278%). The results demonstrate that Sem-MoE can obtain certain performance gains by scheduling the requests across different DP ranks and co-placing the experts in appropriate devices.

**Attention-TP Scenario.** For the attention-TP scenario, there is no scheduler for a single inference instance, as different TP ranks receive the same input. Meanwhile, latency becomes a main concern in TP settings. Therefore, we set the request rate to 1 req/s and vary the input sequence length to observe the TTFT and end-to-end latency directly as shown in Table 2. For Deepseek-V2-Lite, Sem-MoE outperforms the best baseline in TTFT by 12.21%, 10.60%, and 18.89% under input lengths of 256, 512, and 1024. For Qwen3-30B-A3B, the performance optimization ratio is 17.16%, 24.90%, 3.80%. Thanks to the load-balance and inter-layer communication optimizing effect, MoETuner gains performance improvement in some cases, yet may slow down in the other cases. Model-data collaborative scheduling can bring holistic performance boosting in all tested scenarios. The speedup of TTFT also translates to the shrinking of end-to-end inference latency, just as shown in Table 2. We would delve into the execution of MoE layers to analyze the rationale behind the breakdown of the inference speedup.

**Extensive Experiments.** We also conduct performance evaluation using one additional model (Moonlight-16B Liu et al. (2025)), and the results are consistent with the above observations, which are provided in § C.2.

## 4.5 A DETAILED LOOK AT EP COMMUNICATION REDUCTION

In Figure 5(a), we show the local activation rate and the resulting latency of a single MoE layer under the attention-TP scenario. Local activation means the tokens' activation is routed to an expert collocated on the same GPU device, and thus, remote routing and its associated EP communication can be skipped. Results show that, compared with the vanilla placement, Sem-MoE can increase LAR by 37% and 43% for DeepSeek-V2-Lite and Qwen3-30B-A3B, which translates to 41.8%/46.6% latency reduction of the belonging expert layer. Besides Vanilla and Sem-MoE, other bars in the figure are measured by mocking the routing module of SGLang and skipping the delays in communication to fabricate hypothetical baselines just for reference. Note that a 100% LAR may not be achieved in theory, as different tokens can contradict each other to group their own hot experts, but GPU memory is limited.

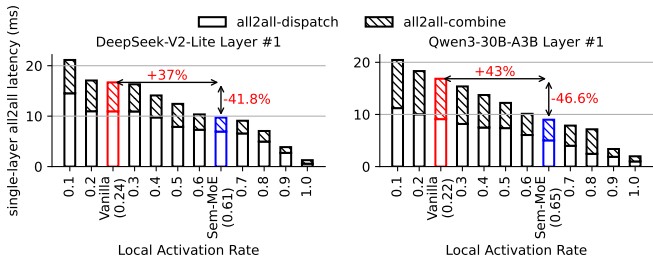

(a) Local activation rate against overall EP overhead.

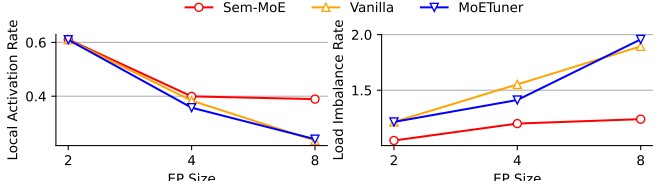

(b) Local activation rate and load-imbalance rate of Sem-MoE and baselines.

Figure 5: Breakdown Evaluation

### 4.6 ALGORITHM EVALUATION

The model-data collaborative scheduling algorithm needs to find balanced co-clusters of tokens and experts, with experts having a maximal likelihood of being gated (routed) from tokens within the same cluster, and minimal likelihood across clusters. An additional regularizer is load balancing that ensures hot and cold experts are relatively evenly distributed. Sem-MoE adopts Algorithm 1 to approximately solve the problem and is evaluated against two baselines, the vanilla scheduling policy (original expert placement policy and round-robin scheduling) and MoETuner. Figure 5(b) shows the averaged local activation rate (ratio of tokens computed at the local device) and load imbalance rate (maximum load divided by the median load) of all the MoE layers in DeepSeek-V2-Lite. Sem-MoE achieves the best local activation rate with balanced expert clusters outperforming the best baseline by 15.4% and 36.7% under EP8 setting.

We also evaluate the cross-dataset zero-shot transfer performance of the predictor model, proving the robustness of the model-data affinity and the generalizability of the scheduling policy, which can be referred to § C.1.

## 5 CONCLUSION

The communication overhead of expert parallelism renders a significant bottleneck in serving large-scale MoE models. We present semantic parallelism and Sem-MoE, which can proactively and losslessly trim EP's `all2all` communication volume via model-data collaborative scheduling, leveraging the intrinsic affinity between model experts and input tokens. Experiments show that Sem-MoE can significantly reduce communication overhead and boost inference throughput under differently specified SLO constraints, both in attention-DP and attention-TP scenarios.

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

## A   EMPIRICAL STUDY OF TOKEN-EXPERT AFFINITY

### A.1   IMPACT OF REQUEST SEMANTICS ON EXPERT ACTIVATION

To better illustrate the impact of request semantics on expert activation, we selected requests from several different types of topics in the MMLU dataset and profiled the expert activations in the 24th layer of Qwen3-30B-A3B, as shown in Figure 6(a). After performing t-SNE dimensionality reduction, it can be observed that requests from similar topics exhibit similarity in activated experts. Requests from the math-related topics of abstract algebra and college mathematics activate similar experts, whereas requests from humanities topics, such as philosophy and professional law, are relatively distant from the math-related ones in the reduced-dimensional space. Sem-MoE leverages this semantic affinity between requests to perform co-scheduling of data and models for requests under the Attention-DP scenario.

### A.2   TOKEN-EXPERT CONJUGACY

**Intra-layer token-expert affinity**   Within each MoE layer, each expert module in an LLM layer is trained to process a particular semantic domain of tokens. Tokens and experts exhibit high affinity in different dimensions. We profile the intermediate activation of the gating module in each MoE layer in the DeepSeek-V2-Lite. One important observation shows that strong bi-clustered conjugacy between tokens and experts, as shown in Figure 6. That is, experts are likely to be activated by a certain sub-group of tokens with high semantic affinity, while they are not likely to be activated by other general tokens in the vocabulary. And from the tokens' perspective, it is true that semantically similar tokens are likely to activate a certain sub-group of experts. This is the preliminary motivation for model-data collaborative scheduling.

**Inter-layer expert-expert affinity**    The left picture of Figure 6(c) shows the activation correlation of the 4th and 5th layer of the Mixtral-8x7B model. The x-/y-axis represents the expert groups of a layer. For Mixtral-8x7B, each token is routed to 2 out of 8 experts at each layer. Thus, the number of expert groups at each layer is $\binom{8}{2} = 28$. When tokens choose some concrete experts at the fourth layer, they tend to choose a rather fixed set of experts at the next layer with high probability. We name this phenomenon *inter-layer expert-expert affinity*.

**Simple conditional probability model for token activation path**    The above examples illustrate the tabularized relationship between tokens and experts. We argue that, simple conditional probability model can work to predict the activation path of tokens by combining the above intra-layer conjugacy and/inter-layer affinity. We first calculate the kurtosis[3] of each token's activation map. The right picture of Figure 6(b) shows most of the kurtosis values are higher than 8, indicating that the to-route experts for each token concentrate on a narrowed set, regardless of the context. We further use partial (25%) of the profile dataset to calculate each tokens' most routed top-k experts. Then the static top-k experts are used to predict the tokens activation using the left part (75%) of the profile dataset, achieving a 96.3% precision and a 78.8% F1 score. The right part of Figure 6(c) predicts the next-layer-activation via looking back the prior layers. As we know more about the previous activation sequence at layer $L$, we can predict the activation at layer $L + 1$ with higher confidence (about 70% when looking back 5 layers).

**Expert pre-grouping and token re-batching**    Leveraging the intrinsic conjugacy between tokens and experts in MoE models to achieve token-expert co-dispatching, may help reduce communication volume and boost distributed parallel inference. First, similar experts can be pre-grouped together at deployment time based on pre-profiled and modeled affinity. Second, on the fly, individual tokens can be re-shuffled and re-batched to the GPU devices that host the expert groups whose member experts have largest modeled conjugacy with the token. Such co-scheduling aims to avoid scattered, cross-device token-expert activations, or equivalently, maximize the probability of intra-device, local activations. Figure 6(d) shows an micro-benchmark experiment, testing the `all2all` latency under different local activation rate using the *nccl* `all2allv` API. With the local activation rate ($\alpha$) varying from 0.2 to 0.9, the latency decreases gradually, showing the performance gains with improved expert-token co-scheduling.

## B    ALGORITHM FOR THE MODEL-DATA CO-SCHEDULING SOLVER

Algorithm 1 describes the model-data co-scheduling alternating optimization algorithm in Sem-MoE. The algorithm achieves optimal performance by alternating between optimizing the scheduling of requests and the placement of experts. Initially, requests are clustered based on their affinity to experts to determine their scheduling (line 44). In each iteration, the algorithm alternates between optimizing expert placement with fixed requests and request scheduling with fixed experts. For optimizing expert placement, experts are first sorted by their hotness in descending order. Given the current request scheduling and expert placement, the affinities between experts and the cluster's experts/requests are computed and aggregated via weights $\alpha_e$ and $\beta_e$, which is adjusted by the cluster's current load to derive a final affinity score (lines 11-13). The expert is assigned to the highest-scoring cluster, with saturated clusters masked (line 14). The algorithm then performs $ft\_steps$ fine-tuning rounds, randomly selects two clusters and swaps their experts if it improves the affinity score (lines 20-25). Request scheduling optimization is similar to expert placement. The req-req affinity and req-expert affinity for each cluster are calculated, aggregated to obtain an affinity score, and the request is scheduled to the cluster with the highest score (lines 28-42).

By now, the token-device scheduling table $\mathcal{T}$, token-device scheduling confidence table $\mathcal{T}_p$, and expert-device scheduling table $\mathcal{E}$ are generated. After the scheduling table $\mathcal{E}$ is constructed, the experts at each layer need to be rearranged according to the scheduling table during online inference service deployment. In addition, the Sem-MoE rearranges the gating module by column to implement transparent expert shuffle. The semantics of other layers are not affected. The rearranged experts are highly boxed, so that the token activation at each layer is de-cohesive, and the redundant network communication overhead caused by dispersive activation is reduced.

---

[3]Kurtosis is a measure of the tailedness of a distribution. High Kurtosis indicates a token favors several fixed experts during multiple occurrences.

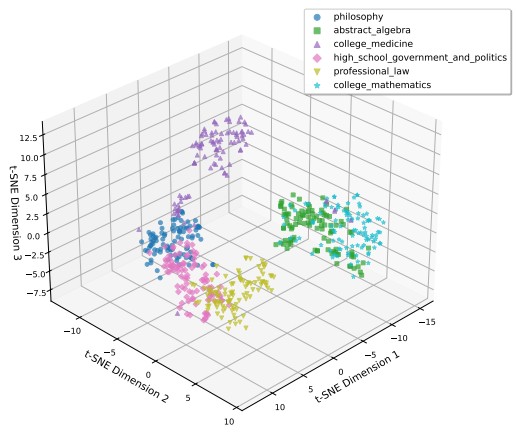

(a) Example of expert activations for requests in different topics (24th layer of Qwen3-30B-A3B profiled using the MMLU dataset), with t-SNE dimensionality reduction.

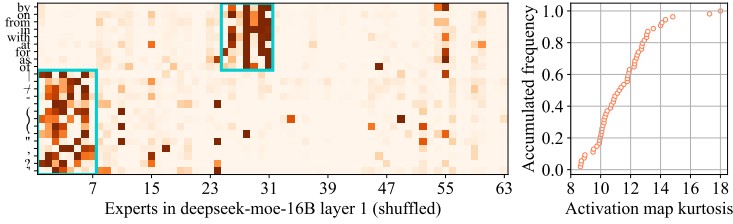

(b) Example of intra-layer token-expert activation map (1st MoE layer of DeepSeek-V2-Lite profiled using the Sharegpt dataset). Darker color in the map indicates higher activation frequency or stronger correlation.

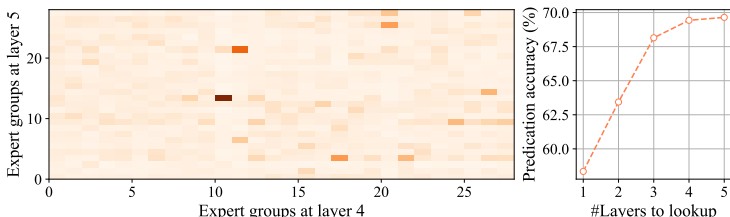

(c) Example of inter-layer expert-expert correlation map (4th/5th MoE layer of Mixtral-8x7B profiled using the LongBench dataset). Darker color in the map indicates stronger correlation.

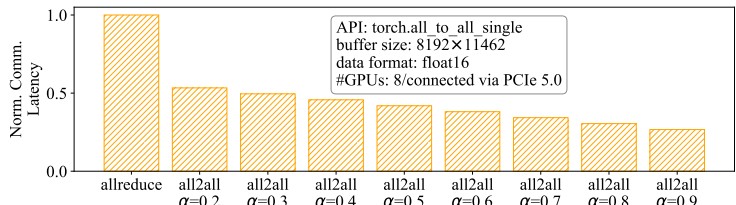

(d) Example of perfromance of `allreduce` and `all2all` under different local activation ratio ($\alpha$).

Figure 6: Conjugacy illustration and collective communication micro-benchmark.

---

**Algorithm 1:** Alternating-based data-model co-scheduling algorithm

---

**input:** $n\_steps$: number of iteration steps;
$\mathcal{C}_p$: the token-2-expert confidence table; $a$: the token frequency;
$r$: requests list; $K$: number of requests;
$N$: number of experts per layer; $E$: number of co-clusters;
$t$: number of tokens;
**output:** $\mathcal{E}$: expert labels; $\mathcal{T}$: token labels;
$\mathcal{T}_p$: confidence of tokens choosing specific experts

1   $p\_matrix\_ep\_opt \leftarrow$ `zeros`$(N, E)$ $/$ $E$
2   $p\_matrix\_req\_opt \leftarrow$ `zeros`$(K, E)$ $/$ $E$
3   **Function** `expert_place`$(\mathcal{C}_p, p\_matrix\_req, \alpha_e, \beta_e)$**:**
4     $loads \leftarrow$ compute per expert load by $(\mathcal{C}_p)$
5     `sort_by_load`$(e, loads)$
6     $mask, cnter \leftarrow$ `ones`$(E)$**,** `zeros`$(E)$
7     $EAfE, EAfR \leftarrow$ `zeros`$(N, E)$**,** `zeros`$(N, E)$
8     $p\_matrix\_ep \leftarrow$ `zeros`$(N, E)$ $/$ $E$
9     $loads\_cls \leftarrow$ `zeros`$(E)$
10    **for** $e$ *in* $e$ **do**
11      $EAfE[e] \leftarrow$ compute expert-expert affinity by $(mask, p\_matrix\_ep, \mathcal{C}_p)$
12      $EAfR[e]$ compute req-expert affinity by $(mask, p\_matrix\_req, \mathcal{C}_p)$
13      $aff\_score \leftarrow \alpha_r * EAfE[e] + \beta_r * EAfR[e] - \gamma_e * loads\_cls$
14      $cls_e \leftarrow \arg\max_{cls} aff\_score$
15      $p\_matrix\_ep[e][cls_e] \leftarrow 1$
16      $cnter[cls_e] \leftarrow cnter[cls_e] + 1$
17      **if** $cnter[cls_e] >= N/E$ **then**
18       $maks[cls_e] \leftarrow 0$
19      $loads\_cls[cls_e] \leftarrow loads\_cls[cls_e] + loads[e]$
20    **repeat**
21      $cls_1, cls_2 \leftarrow$ randomly select a cluster in $[\![E]\!]$
22      $e_1, e_2 \leftarrow$ randomly select experts in $p\_matrix\_ep[:][cls_1]$ and $p\_matrix\_ep[:][cls_2]$
23      **if** `aff_gain`$(e_1, e_2, cls_1, cls_2) > 0$ **then**
24       `swap`$(e_1, e_2, p\_matrix\_ep)$
25    **until** *iterating for ft_steps steps*
26    **return** $p\_matrix\_eq$

27

28 **Function** `request_schedule`$(\mathcal{C}_p, p\_matrix\_ep, \alpha_r, \beta_r)$**:**
29    `sort_by_len`$(r)$
30    $mask, cnter \leftarrow$ `ones`$(E)$**,** `zeros`$(E)$
31    $RAfR, RAfE \leftarrow$ `zeros`$(K, E)$**,** `zeros`$(K, E)$
32    $p\_matrix\_req \leftarrow$ `zeros`$(K, E)$ $/$ $E$
33    **for** $r$ *in* $r$ **do**
34      $RAfR[r] \leftarrow$ compute req-req affinity by $(mask, p\_matrix\_req, \mathcal{C}_p)$
35      $RAfE[r] \leftarrow$ compute req-expert affinity by $(mask, p\_matrix\_ep, \mathcal{C}_p)$
36      $aff\_score \leftarrow \alpha_r * RAfR[r] + \beta_r * RAfE[r]$
37      $cls_r \leftarrow \arg\max_{cls} aff\_score$
38      $p\_matrix\_req[r][cls_r] \leftarrow 1$
39      $cnter[cls_r] \leftarrow cnter[cls_r] + 1$
40      **if** $cnter[cls_r] >= K/E$ **then**
41       $maks[cls_r] \leftarrow 0$
42    **return** $p\_matrix\_req$

43

44 $p\_matrix\_req \leftarrow$ cluster based on expert affinity
45 **repeat**
46    $p\_matrix\_ep \leftarrow$ `expert_place`$(\mathcal{C}_p, p\_matrix\_req, \alpha_e, \beta_e)$
47    $p\_matrix\_req \leftarrow$ `request_schedule`$(\mathcal{C}_p, p\_matrix\_ep, \alpha_r, \beta_r)$
48    $scores \leftarrow$ summation the max load and communication cost given $p\_matrix\_eq$ and $p\_matrix\_req$
49    $better\_scheduling \leftarrow$ samples with scores
50    update $p\_matrix\_ep\_opt$ and $p\_matrix\_req\_opt$
51 **until** *iterating for n_steps steps*

52

53 $\mathcal{E} \leftarrow$ `argmax`$(p\_matrix\_ep\_opt, axis{=}1)$
54 $p\_matrix\_tk\_opt \leftarrow$ count the tokens per req in $p\_matrix\_req\_opt$
55 $\mathcal{T}, \mathcal{T}_p \leftarrow$ `argmax_with_values`$(p\_matrix\_tk\_opt, axis{=}1)$

---

## B.1 Modeling Inter-layer Activation Conjugacy

Leveraging the conditional probability model described in § A.2, we use a simple probability-based first-order Marcov chain to model the inter-layer activation conjugacy. To reduce the combination space, we model the activation device sequence rather than the activation expert sequence, because we only care about the device-level token rebatching. When looking back $l$ layers, we construct a table shaped like $[E^l, E]$, where the row of the table indicates the sequence of devices selected at the previous $l$ layers and the column indicates the probability of activating the $E$ devices in the current layer. Like the § B shows, we also calculate the activation sequence to device table $\mathcal{A}$ and the confidence table $\mathcal{A}_p$. In practice, we set the number of looking-back layers as 2.

---

**Algorithm 2:** Online request scheduling based on fast lookup

**input:** $\mathcal{R} \in \mathbb{N}^n$: Input requests; $\mathcal{T}$: token-to-expert-cluster Schedule Table; $E$: number of DP size

1  $dev\_mask \leftarrow \text{ones}(E)$
2  **Function** get_dp_rank($\mathcal{R}$, $\mathcal{T}$)**:**
3      $dev\_score \leftarrow \text{sum}(\mathcal{T}[\mathcal{R}, :], dim = 0)$
4      $dev\_score[\,dev\_mask] \leftarrow -\inf$
5      $dev\_id \leftarrow \text{argmax}(dev\_score)$
6      $dev\_mask[dev\_id] \leftarrow False$
7      **if** $dev\_mask$ all are False **then**
8          $dev\_mask \leftarrow \text{ones}(E)$
9      **return** $dev\_id$
10 $dev\_id \leftarrow$ get_dp_rank($\mathcal{R}$, $\mathcal{T}$)
11 schedule($\mathcal{R}$, $dev\_id$)

---

## B.2 Speculative Token Shuffling on the Fly Based on Fast Lookup

To reduce the combination space, we model the activation device sequence rather than the activation expert sequence, because we only care about the device-level token rebatching. We implement a fast online token re-batching mechanism based on fast looking-up tables in both Attention-DP and Attention-TP (Algorithm 2 & Algorithm 3).

**Data Scheduling: Attention-DP Scenarios.** The algorithm 2 queries the token-to-expert-cluster scheduling table $\mathcal{T}$ based on the token IDs appearing in the request $\mathcal{R}$, and aggregates the results to obtain a score for each device for that request (line 3). Then $\mathcal{R}$ is scheduled to the device with the max valid score (line 5). To prevent requests biased toward a subset of experts, which could skew the load during the decoding phase, we introduce a $dev\_mask$. The device is masked after it is allocated (line 4-5). Once a round of allocation is completed and all devices are masked, the $dev\_mask$ is reset and enters a new round (line 7-8). This ensures that Sem-MoEachieves expert affinity while maintaining load balance across devices.

**Data Scheduling: Attention-TP Scenarios.** The algorithm 3 queries the token-to-expert-cluster scheduling table $\mathcal{T}$ and expert-cluster-sequence-to-expert-cluster table $\mathcal{S}$, together with their confidences first. Then, the table with higher confidence is adopted to obtain the device ID list to which the current batch token needs to be shuffle (line 2). The algorithm performs the argsort operation to obtain the shuffle indicators (line 3) of the token. Then, the final shuffle indicators are obtained by grouping, aligning, and concatenation, and the token is shuffled (line 4 to line 7). After rebatching is complete, Sem-MoE calls the `reduce-scatter` operation. After MoE computing is complete, Sem-MoE runs the `allgather` operation to collect tokens. Finally, the order of tokens are shuffled back based on the previously calculated shuffle indicators (lines 14-18).

The both algorithm do not involve complex load calculation and decision-making. They are directly completed by querying tables. The runtime overhead mainly involves large token matrix shuffling, which we optimize via high-performance kernels. The memory occupation of the scheduling tables is negligible. For example, for DeepSeek-V2, the memory space that the token-to-device table $\mathcal{T}$ occupies is $\frac{102400 \times 60 \times 2}{1024^2} \approx 11.72 MB$ (assuming the data format is `int16`).

---

**Algorithm 3:** Online token re-batching based on fast lookup

---

**input:** $\mathcal{B} \in \mathbb{N}^n$: Input token IDs; $\mathcal{T}$: token-to-expert-cluster Schedule Table;
$\mathcal{A}$: expert-cluster-sequence-to-expert-cluster Schedule Table

1 **Function** rebatch_tokens($\mathcal{B}, \mathcal{T}$):
2    $dev\_ids \leftarrow$ cond($\mathcal{T}_p[\mathcal{B}] > \mathcal{A}_p[\mathcal{B}], \mathcal{T}[\mathcal{B}], \mathcal{A}[\mathcal{B}]$)
3    $shf\_indices \leftarrow$ argsort($dev\_ids$)
4    $g\_shf\_indices \leftarrow$ group_by_key($shf\_indices$)
5    $g\_shf\_indices \leftarrow$ align($g\_shf\_indices$)
6    $shf\_indices \leftarrow$ concat($g\_shf\_indices$)
7    $\mathcal{B} \leftarrow \mathcal{B}[shf\_indices]$
8    **return** $shf\_indices$

9
10 **Function** resume_tokens($\mathcal{B}, shf\_indices$):
11    $r\_shf\_indices \leftarrow$ argsort($shf\_indices$)
12    $\mathcal{B} \leftarrow \mathcal{B}[r\_shf\_indices]$

13
14 $shf\_indices \leftarrow$ rebatch_tokens($\mathcal{B}, \mathcal{T}$)
15 $\mathcal{B}_{local} \leftarrow$ reduce_scatter($\mathcal{B}$)
16 executing MoE layer
17 $\mathcal{B} \leftarrow$ allgather($\mathcal{B}_{local}$)
18 resume_tokens($\mathcal{B}, shf\_indices$)

---

## C EXTENSIVE EXPERIMENTS

### C.1 CROSS-DATASET VALIDATION OF EXPERT ROUTING PREDICTION

To assess the generalization of our method, we conducted cross-domain experiments using two large language models: DeepSeek-V2-Lite and Qwen3-30B. Specifically, we trained Sem-MoE's activation predictor on a single source dataset and evaluated the quality of its token-expert co-scheduling decisions, measured by the Local Activation Rate (LAR), on the other two unseen, out-of-distribution target datasets. Note that LAR measures the proportion of tokens routed to local experts, reflecting a reduced communication volume to experts on remote devices. Therefore, a higher LAR indicates lower cross-device communication overhead—and is better.

The results reveal robust zero-shot transfer performance for our method.

Table 3: Cross-dataset evaluation of zero-shot transfer for DeepSeek-V2-Lite

| LAR (p50) Method | ShareGPT | Lmsys-Chat-1M | MMLU |
|---|---|---|---|
| Sem-MoE trained on ShareGPT | **46.49%** | 41.25% | 41.55% |
| Sem-MoE trained on Lmsys-Chat-1m | 40.68% | **47.19%** | 41.40% |
| Baseline: SGLang | 24.98% | 25.01% | 24.97% |

Table 4: Cross-dataset evaluation of zero-shot transfer for Qwen3-30B

| LAR (p50) Method | ShareGPT | Lmsys-Chat-1M | MMLU |
|---|---|---|---|
| Sem-MoE trained on ShareGPT | **46.88%** | 39.28% | 35.30% |
| Sem-MoE trained on Lmsys-Chat-1m | 36.57% | **43.61%** | 33.37% |
| Baseline: SGLang | 25.00% | 25.01% | 25.00% |

As shown in Table 3, for DeepSeek-V2-Lite, the predictor trained on ShareGPT achieves an in-distribution LAR of 46.49%. When applied without retraining or fine-tuning to Lmsys-Chat-1M, it still attains 41.25% LAR—only a modest drop from the best in-domain result on that dataset (47.19%, achieved when training directly on Lmsys-Chat-1M). More importantly, this out-of-distribution performance is 1.65× higher than the SGLang's default scheduling setting (25.01%). Similarly, on

MMLU—a domain with very different content and structure—the same ShareGPT-trained predictor yields 41.55% LAR, far surpassing SGLANG defaults (24.97%) and remaining close to in-distribution levels.

As shown in Table 4, the trend is consistent for Qwen3-30B: the ShareGPT-trained predictor achieves 39.28% LAR on Lmsys-Chat-1M and 35.30% on MMLU, compared to peak in-domain scores of 43.61% and 36.57% (when trained on Lmsys), and both are substantially above the SGLang baseline ( 25%). A similar trend holds when transferring reservedly from Lmsys-Chat-1M to ShareGPT and MMLU.

These results demonstrate that when fed with real-world, representative datasets, Sem-MoE tends to capture the invariability in token-routing patterns that remain effective across different linguistic styles, task types, and knowledge domains. While training on the target distribution offers marginal improvements, it is not required except for extreme performance pursuits.

## C.2 THROUGHPUT IMPROVEMENT OF MOONLIGHT-16B

To address the issue of model diversity, we evaluate the Moonlight-16B model. The results align closely with the performance trends observed for DeepSeek-V2-Lite and Qwen3-A30B, as reported in the § 4. In summary, our approach Sem-MoE, consistently outperforms SGLANG (MoETuner): it achieves an average performance improvement of 1.12× (1.22×) for prefill and 1.10× (1.14×) for end-to-end inference, respectively. Detailed results are provided in Table 5 and Table 6.

Table 5: Throughput Optimization Effect under TTFT SLO

| Throughput (Req/s) Dataset | SGLang | MoETuner | Sem-MoE | Speedup v.s. SGLang | Speedup v.s. MoETuner |
|---|---|---|---|---|---|
| Lmsys-Chat-1M | 32.7 | 28.6 | **38.6** | 1.18x | 1.35x |
| MMLU | 32.2 | 30.1 | **38.6** | 1.20x | 1.28x |
| ShareGPT | 2.9 | 2.8 | **2.9** | 1.00x | 1.02x |
| Average | 22.6 | 20.5 | **26.7** | 1.12x | 1.22x |

Table 6: Throughput Optimization Effect under E2E latency SLO

| Throughput (Req/s) Dataset | SGLang | MoETuner | Sem-MoE | Speedup v.s. SGLang | Speedup v.s. MoETuner |
|---|---|---|---|---|---|
| Lmsys-Chat-1M | 37.2 | 35.5 | **38.6** | 1.04x | 1.09x |
| MMLU | 37.9 | 35.3 | **38.6** | 1.02x | 1.10x |
| ShareGPT | 1.9 | 1.9 | **2.4** | 1.23x | 1.23x |
| Average | 25.7 | 24.2 | **26.5** | 1.10x | 1.14x |

