# OpenReview forum: "Semantic Parallelism: Redefining Efficient MoE Inference via Model-Data Co-Scheduling"
_ICLR.cc/2026/Conference — ICLR 2026 Poster_

### Official Review · Reviewer_4HBM · 2025-10-29

**Soundness:** 2
**Presentation:** 3
**Contribution:** 3
**Rating:** 4
**Confidence:** 4

**Summary:**

This paper introduces Sem-MoE, a mixture-of-experts (MoE) scheduling framework that leverages offline profiling of token–expert activations to optimize communication and load balance. The method first computes a token–expert activation matrix $$C_{p,jk}$$ from model logs, then formulates an integer linear program (ILP) minimizing cross-expert communication and balancing expert loads. The ILP solution is later applied at runtime (§3.4) to pre-group experts and improve all-to-all efficiency. Experiments on an 8×A100 NVLink setup report a 2.78× throughput gain and 25% latency reduction compared with FasterMoE.

 The overall mathematical formulation and implementation are coherent and elegant, yet the theoretical and empirical justifications for using offline token–expert frequencies as stable, context-independent features are insufficient. Experimental validation focuses mainly on efficiency and omits semantic and multi-node evaluation.

**Strengths:**

(1) The ILP modeling in §3.3 is mathematically elegant and well aligned with the MoE scheduling problem. It directly encodes both communication minimization and load balancing as discrete constraints. The objective function depends explicitly on $$C_{p,jk}$$, linking statistical observations to optimization variables. The derivation is concise and solvable, and the implementation (§3.4) confirms that this optimization is executed in practice.

(2) The pipeline from offline profiling to runtime integration is clear and systematic. The ILP-generated token–expert and expert–expert tables are concretely used for expert grouping, confirming that the method bridges theory and system.

(3) The experiments (§4) show consistent and interpretable performance gains, demonstrating real engineering improvement, not mere theoretical potential.

**Weaknesses:**

(1) The ILP depends on the token–expert frequency matrix $$C_{p,jk}$$, which the authors assume to be stable and context-independent (§3.2). However, this assumption lacks theoretical grounding or rigorous validation. Appendix A reports stability (96.3%) and F1 (78.8%) computed on a single corpus (ShareGPT) and layer, but provides no variance analysis across datasets, seeds, or layers. Since the gating probability $$P(E_k|x_j,h_{L,j})$$ inherently depends on contextual hidden states $$h_{L,j}$$, averaging over contexts removes critical variance information. Moreover, although the authors describe this as an “offline distribution,” the collection of $$C_{p,jk}$$ itself is derived from attention representations obtained during **normal Transformer model inference on full sentences**. Thus, each token’s recorded expert activation already embeds contextual dependencies from its neighboring tokens. This makes the so-called offline mapping not truly context-independent but an average over context-conditioned activations. Consequently, the statistical input to the ILP may not reflect real routing behavior when context distributions shift.

(2) Figures 6(b)(c) mainly show high-frequency function tokens such as “the”, “and”, “you”. Their activation stability is trivial and does **not** generalize to some specific semantic or rare tokens, which dominate real workloads. Furthermore, Figure 6 shows that activation entropy increases in deeper layers, yet the paper interprets this as “uniform expert reuse.” Uniformity here indicates uncertainty, not stability, thus contradicting the claimed “context independence.” The analysis therefore captures lexical regularity rather than semantic invariance.

(3) The communication cost term in the ILP objective aggregates token-level co-activation frequencies into expert–expert communication weights (§3.3), but no strong empirical correlation between these weights and actual communication traffic is shown. The paper lacks ablation studies testing ILP sensitivity to noise or sampling bias in $$C_{p,jk}$$.

(4) The reported results measure only throughput, latency, and load variance, **lacking evaluation** of model quality (perplexity, accuracy, downstream metrics). So it is unclear whether the offline routing preserves semantic fidelity or harms inference correctness.

(5) All experiments are performed on a single 8×A100 NVLink node (§4). The paper suggests that the method could achieve larger gains under slower interconnects (PCIe or InfiniBand), but this claim remains untested. Including cross-node or heterogeneous-network experiments would strengthen the scalability argument.

**Questions:**

(1) The paper defines “context-independent routing” (§3.2) based on the observed stability of token–expert activation frequencies computed from full-sentence attention representations. However, the gating function $$G_L(h_{L,j})$$ inherently depends on contextual hidden states. How do the authors justify that the offline probability $$P(E_k|x_j)$$ estimated under contextualized inputs approximates the true context-independent term $$P(E_k|x_j,h_{L,j})$$? In particular, what does “semantic independence” mean in this formulation—is it statistical stability of frequent tokens, or an actual invariance of routing under different semantic contexts?

(2) The authors report stability (96.3%) and F1 (78.8%) on ShareGPT, but the analysis seems limited to a single layer and dataset. Could they clarify whether similar stability holds across different layers, seeds, or corpora?

(3) Figures 6(b)(c) analyze only common function tokens. The text briefly mentions that rare tokens are “more uniformly distributed,” but no quantitative results are given. Could the authors elaborate on the behavior of low-frequency or semantic tokens?

(4) The author mentioned “the benefit will be even larger under slower interconnects (PCIe or InfiniBand)” in §4 and suggests that greater improvements would be observed under slower interconnects (PCIe or InfiniBand), yet this claim is untested. Have the authors evaluated or simulated multi-node scenarios to confirm that the ILP-based grouping remains effective when communication latency and bandwidth vary?

---

> ### Author Response · Authors · 2025-11-20
> **Response to Reviewer 4HBM**
>
> We sincerely thank Reviewer 4HBM for your thoughtful and constructive feedback. We’re encouraged by your positive remarks on the coherence and elegance of our mathematical formulation and system implementation, and we appreciate your suggestions for further strengthening the paper. Below, we address each of your concerns in detail.
>
> *In addition, we kindly request you to consider raising the score if our clarifications have adequately addressed your concerns.*

---

> ### Author Response · Authors · 2025-11-20
> **Response to Reviewer 4HBM Weakness #1 (Part 1)**
>
> ### [Reviewer 4HBM-W1] :
>
> > The ILP depends on the token–expert frequency matrix $$C_{p,jk}$$, which the authors assume to be stable and context-independent (§3.2). However, this assumption lacks theoretical grounding or rigorous validation.
> >
>
> > Thus, each token’s recorded expert activation already embeds contextual dependencies from its neighboring tokens. This makes the so-called offline mapping not truly context-independent but an average over context-conditioned activations.
> >
>
> ### [Response to 4HBM-W1] Clarification and in-depth analysis of  on “context-independence”
>
> We want to clarify that it would be inaccurate to claim expert selection is absolutely independent of a token’s context (i.e., the prompt sentence).  However, extensive evidence shows that expert routing decisions for a large fraction tokens remain highly stable across diverse prompts. **Rather than treating “context-independence” as a binary property, we view it as a spectrum**. Thus, the **practical invariability in routing** is exploitable—Sem-MoE leverages this for real system gains without overstating the phenomenon.
>
> **Prior study.** Prior work [1] identified *context-independent specialization*: certain experts consistently handle specific tokens regardless of a token's surrounding text; furthermore, the paper investigates and claims that this stability arises because token-to-expert mappings are largely fixed early in pre-training.
>
> [1] Xue, Fuzhao, et al. "OpenMoE: an early effort on open mixture-of-experts language models." ICML 2024.
>
> **Our more detailed quantitative analysis and support on  “context-independence”**
>
> - Our Sem-MoE method remains highly effective even with input tokens exhibiting extremely diverse contextual conditions. As shown in Tables 1 and 2, we analyze token-expert activation across over 20k distinct tokens—spanning 441K and 217K contextual instances in ShareGPT and LMSYS-Chat-1M datasets, respectively. Table1 lists the top-5 most frequent tokens (with and without stop words), each appearing in 630–30k unique contexts; Table2 extends this to tail tokens covering 70% of occurrences, which also show high contextual diversity. Despite every tokens with a minimal proportion of unique contexts reaching 95.8%, Sem-MoE maintains consistent routing patterns, reducing unnecessary communication and achieving up to 24.9% lower latency (2.78× higher throughput).
> - We report the **variance—IQR (inter-quantile range or P25, P50, P70 quantiles)—of F1-score** for Sem-MoE’s token-routing predictor for DeepSeek-V2-Lite and Qwen3-30B and on datasets ShareGPT and LMSYS-Chat-1M.  Experiments confirm that **token-to-expert activation can be accurately predicted across layers, models and datasets**. For DeepSeek-V2-Lite, the median (p50) token achieves high F1 scores (0.833–1.0) on 80% of the layers (21 out of 26) in predicting its 6 activated experts out of 64 on both datasets. Qwen3-30B shows a similar trend: 80% and 60% of layers exceed a 0.75 median F1-score when predicting 8 of 128 experts on the two datasets—a more challenging task. At the upper end, p75 F1-scores are often near-perfect (~1.0); at the lower end, mean p25 F1-scores are 0.667/0.654 for DeepSeek-V2-Lite and 0.547/0.521 for Qwen3 on the datasets. (See DeepSeek-V2-Lite on ShareGPT in Table3, others omitted due to limited space)
> - For any given token, expert activation across diverse contexts is highly skewed and stable: the token consistently routes to a small, fixed subset of “top-k experts,” while the remaining experts stay largely inactive (“cold”). Across all layers of DeepSeek-V2-Lite and Qwen3 on both datasets, the median (p50) cumulative hotness of top-k experts ranges from 0.833 to 0.976, whereas the maximum hotness among cold experts remains low—around 0.05 at p50—demonstrating a consistent and pronounced activation disparity. (See DeepSeek-V2-Lite on ShareGPT in Table3, others omitted due to limited space)
>
> Overall, **we do NOT claim universal context-independence—only that observed routing invariance is strong enough to be practically useful**. Sem-MoE capitalizes on this regularity to deliver significant system-level gains without overreliance on idealized assumptions. The phenomenon is real, measurable, and exploitable—even if imperfect.

---

> ### Author Response · Authors · 2025-11-20
> **Response to Reviewer 4HBM Weakness #1 (Part 2)**
>
> Table 1: Context diversity for the hottest 10 words (w/ and w/o stop-words)
>
> | Token | Hotness Rank | Unique contexts | Num. occurrences | Proportion of uniques | Token (w/o stopwords) | Hotness Rank | Unique num. contexts | Num. occurrences | Proportion of uniques |
> | --- | --- | --- | --- | --- | --- | --- | --- | --- | --- |
> | Ċ | 1 | 30150 | 30246 | 0.997 | \\_ | 1 | 1720 | 1720 | 1.000 |
> | . | 2 | 15518 | 15629 | 0.993 | _\\ | 2 | 683 | 683 | 1.000 |
> | Ġthe | 3 | 12073 | 12171 | 0.992 | âĢĻ | 3 | 670 | 676 | 0.991 |
> | , | 4 | 11939 | 12095 | 0.987 | Ġdata | 4 | 662 | 665 | 0.995 |
> | Ġand | 5 | 7566 | 7641 | 0.990 | Ġuse | 5 | 630 | 632 | 0.997 |
>
> &nbsp;
> &nbsp;
> &nbsp;
> &nbsp;
> &nbsp;
> &nbsp;
>
> Table 2: Context diversity for the hottest words covering 70% overall token occurrences  (w/ and w/o stop-words)
>
> | Token | Hotness Rank | Unique contexts | Num. occurrences | Proportion of uniques | Token (w/o stopwords) | Hotness Rank | Unique num. contexts | Num. occurrences | Proportion of uniques |
> | --- | --- | --- | --- | --- | --- | --- | --- | --- | --- |
> | Ċ | 1 | 30150 | 30246 | 0.997 | \\_ | 1 | 1720 | 1720 | 1.000 |
> | Ġthey | 91 | 430 | 432 | 0.995 | Ġcharacters | 386 | 71 | 71 | 1.000 |
> | Ġvalue | 181 | 218 | 218 | 1.000 | Ġcontinue | 771 | 45 | 45 | 1.000 |
> | ') | 271 | 143 | 143 | 1.000 | Ġtalk | 1156 | 32 | 32 | 1.000 |
> | h | 361 | 111 | 111 | 1.000 | Search | 1541 | 26 | 26 | 1.000 |
> | '); | 451 | 93 | 93 | 1.000 | Ġcontrols | 1926 | 21 | 21 | 1.000 |
> | Ġquality | 541 | 79 | 79 | 1.000 | ĠSuccess | 2311 | 18 | 18 | 1.000 |
> | Ġyear | 631 | 64 | 69 | 0.928 | Ġencourage | 2696 | 15 | 15 | 1.000 |
> | Ġarticle | 721 | 61 | 62 | 0.984 | language | 3081 | 14 | 14 | 1.000 |
> | end | 811 | 55 | 56 | 0.982 | Ġsuggested | 3466 | 12 | 12 | 1.000 |
> | Ġhave | 883 | 51 | 53 | 0.962 | Ġokay | 3774 | 11 | 11 | 1.000 |
>
> &nbsp;
> &nbsp;
> &nbsp;
> &nbsp;
> &nbsp;
> &nbsp;
>
> Table 3: Context-independent analysis for all layers of DeepSeek-V2-Lite on  ShareGPT
>
> | Layer ID | F1-score p25 | F1-score p50 | F1-score p75 | Cum. hotness for Top-k frequent experts p25 | Cum. hotness for Top-k frequent experts p50 | Cum. hotness for Top-k frequent experts  p75 | Maximal Hotness for Non-top-k Experts p25  | Maximal Hotness for Non-top-k Experts p50 | Maximal Hotness for Non-top-k Experts p75  |
> | --- | --- | --- | --- | --- | --- | --- | --- | --- | --- |
> | 0 | 0.833 | 1.000 | 1.000 | 0.833 | 0.976 | 1.000 | 0.000 | 0.042 | 0.071 |
> | 1 | 0.667 | 0.833 | 1.000 | 0.722 | 0.917 | 1.000 | 0.000 | 0.048 | 0.071 |
> | 2 | 0.833 | 1.000 | 1.000 | 0.833 | 0.944 | 1.000 | 0.000 | 0.048 | 0.071 |
> | 3 | 0.833 | 1.000 | 1.000 | 0.833 | 0.944 | 1.000 | 0.000 | 0.050 | 0.078 |
> | 4 | 0.667 | 0.833 | 1.000 | 0.778 | 0.917 | 1.000 | 0.000 | 0.051 | 0.083 |
> | 5 | 0.667 | 0.833 | 1.000 | 0.762 | 0.917 | 1.000 | 0.000 | 0.056 | 0.083 |
> | 6 | 0.667 | 0.833 | 1.000 | 0.767 | 0.917 | 1.000 | 0.000 | 0.056 | 0.083 |
> | 7 | 0.667 | 0.833 | 1.000 | 0.733 | 0.917 | 1.000 | 0.000 | 0.056 | 0.083 |
> | 8 | 0.667 | 0.833 | 1.000 | 0.750 | 0.917 | 1.000 | 0.000 | 0.056 | 0.083 |
> | 9 | 0.667 | 0.833 | 1.000 | 0.750 | 0.917 | 1.000 | 0.000 | 0.056 | 0.083 |
> | 10 | 0.667 | 0.833 | 1.000 | 0.722 | 0.889 | 1.000 | 0.000 | 0.056 | 0.083 |
> | 11 | 0.667 | 0.833 | 1.000 | 0.722 | 0.875 | 1.000 | 0.000 | 0.056 | 0.083 |
> | 12 | 0.500 | 0.667 | 1.000 | 0.667 | 0.833 | 1.000 | 0.000 | 0.051 | 0.083 |
> | 13 | 0.667 | 0.833 | 1.000 | 0.727 | 0.889 | 1.000 | 0.000 | 0.051 | 0.083 |
> | 14 | 0.667 | 0.833 | 1.000 | 0.722 | 0.875 | 1.000 | 0.000 | 0.051 | 0.083 |
> | 15 | 0.667 | 0.833 | 1.000 | 0.702 | 0.875 | 1.000 | 0.000 | 0.053 | 0.083 |
> | 16 | 0.667 | 0.833 | 1.000 | 0.750 | 0.900 | 1.000 | 0.000 | 0.051 | 0.083 |
> | 17 | 0.667 | 0.833 | 1.000 | 0.698 | 0.875 | 1.000 | 0.000 | 0.050 | 0.083 |
> | 18 | 0.667 | 0.833 | 1.000 | 0.667 | 0.850 | 1.000 | 0.000 | 0.051 | 0.083 |
> | 19 | 0.500 | 0.667 | 0.833 | 0.639 | 0.833 | 1.000 | 0.000 | 0.050 | 0.083 |
> | 20 | 0.500 | 0.667 | 0.833 | 0.667 | 0.833 | 1.000 | 0.000 | 0.050 | 0.074 |
> | 21 | 0.500 | 0.667 | 0.833 | 0.639 | 0.833 | 1.000 | 0.000 | 0.050 | 0.074 |
> | 22 | 0.667 | 0.833 | 1.000 | 0.625 | 0.833 | 1.000 | 0.000 | 0.050 | 0.083 |
> | 23 | 0.500 | 0.667 | 0.833 | 0.611 | 0.833 | 1.000 | 0.000 | 0.048 | 0.083 |
> | 24 | 0.667 | 0.833 | 1.000 | 0.641 | 0.833 | 1.000 | 0.000 | 0.050 | 0.083 |
> | 25 | 0.667 | 0.833 | 1.000 | 0.667 | 0.833 | 1.000 | 0.000 | 0.051 | 0.083 |

---

> ### Author Response · Authors · 2025-11-20
> **Response to Reviewer 4HBM Weakness #2**
>
> ### [Reviewer 4HBM-W2] :
>
> > Figures 6(b)(c) mainly show high-frequency function tokens such as “the”, “and”, “you”. Their activation stability is trivial and does **not** generalize to some specific semantic or rare tokens, which dominate real workloads.
> >
>
> > The analysis therefore captures lexical regularity rather than semantic invariance.
> >
>
> ### [Response to 4HBM-W2]:**“Context-independence“  becomes more prominent in our analysis for “rare tokens”**, when excluding stop words.
>
>  The conclusions in [4HBM - Weakness #1] remain valid.  The results summarized in Table 4 show “rare tokens” exhibit even more stable and predictable routing patterns compared to the all-token settings (that include highly frequent trivial stop-words).
>
> (a) For DeepSeek-V2-Lite, the median (p50) **F1-score of Sem-MoE’s predictor keeps 1.0 for the first 10 layers** on both ShareGPT and LMSYS-Chat-1 and declines moderately to **0.833–1.0 for the remaining 15 layers**. On the tail, the **p25 F1-scores for all layers are above 0.6 in both** DeepSeek-V2-Lite on the two datasets**.** A similar trend holds for Qwen3 across datasets. The result ****indicates that these “rare tokens” are routed in a more stable and predictable manner.  Intuitively, “rare tokens” typically carry concrete, specialized semantic cores (e.g., technical terms, specific entities)  and are prone to be attended by a narrow spectrum of domain-specific, closely-related tokens during self-attention, whose hidden states are less affected and susceptible across varied broad contexts.
>
> (b) Expert activation also exhibits markedly **higher skewness for individual tokens, across all layers for the two models on the two datasets** . For a token, the p50 cumulative hotness of a dedicated set of experts mostly activated across contexts—top-k experts—consistently lies between 0.833–1.000.** Similarly, for the non-top-k remaining experts, none of their hotness exceeds beyond 0.042 or 0.056 for the two models across all layers on the two datasets.
>
> Overall, (a) and (b) reinforce the validity and practical relevance of “context independence” for “rare tokens” and thereby strengthen our confidence that Sem-MoE’s design aligns with how modern MoE models behave.
>
> Table 4: Context-independent analysis for DeepSeek-V2-Lite on ShareGPT with stop-words excluded
>
> | Layer ID | F1-score p25 | F1-score p50 | F1-score p75 | Cum. hotness for Top-k frequent experts p25 | Cum. hotness for Top-k frequent experts p50 | Cum. hotness for Top-k frequent experts p75 | Maximal Hotness for Non-top-k Experts p25 | Maximal Hotness for Non-top-k Experts p50 | Maximal Hotness for Non-top-k Experts p75 |
> | --- | --- | --- | --- | --- | --- | --- | --- | --- | --- |
> | 0 | 1.000 | 1.000 | 1.000 | 0.833 | 1.000 | 1.000 | 0.000 | 0.042 | 0.071 |
> | 5 | 0.833 | 1.000 | 1.000 | 0.767 | 0.917 | 1.000 | 0.000 | 0.056 | 0.083 |
> | 10 | 0.833 | 1.000 | 1.000 | 0.722 | 0.889 | 1.000 | 0.000 | 0.056 | 0.083 |
> | 15 | 0.667 | 0.833 | 1.000 | 0.708 | 0.889 | 1.000 | 0.000 | 0.053 | 0.083 |
> | 20 | 0.667 | 0.833 | 1.000 | 0.667 | 0.833 | 1.000 | 0.000 | 0.050 | 0.074 |
> | 25 | 0.667 | 0.833 | 1.000 | 0.667 | 0.833 | 1.000 | 0.000 | 0.051 | 0.083 |

---

> ### Author Response · Authors · 2025-11-20
> **Response to Reviewer 4HBM Weakness #3**
>
> ### [Reviewer 4HBM-W3] :
>
> > The communication cost term in the ILP objective aggregates token-level co-activation frequencies into expert–expert communication weights (§3.3), but no strong empirical correlation between these weights and actual communication traffic is shown.
> >
>
> ### [Response to 4HBM-W3]: **The ILP objective well aligns with communication performance both in theory and through real measurement.**
>
> (a) In theory, a higher local activation rate leads to fewer token-expert remote routing and reduces communication traffic in network resulting in lower communication overhead. Designing ILP to minimize remote activation rate (equivalently maximizing local activation rate) is sound for communication reduction.
>
> (b) We experimentally measure the correlation between the second optimal objective term and the real all2all communication latency, as results shown in Table 5. The results show that as the local activation rate increases, the value of the objective term drops, which translates to significant reductions in actual all2all communication time (or reduce network overheads).
>
> These results confirm that our formulated ILP has an effective and rational objective for communication-reduction in MoE inference.
>
> Table 5: Experimenting the correlation between the ILP Objective term and communication acceleration (traffic reduction)
>
> | Local  Activation Rate (Token-Expert Co-Activation Rate) | ILP Objective Term | Real measured all2all communication latency (ms) |
> | --- | --- | --- |
> | 0.0 | 256k | 5.797 |
> | 0.1 | 230.4k | 5.782 |
> | 0.2 | 204.8k | 5.775 |
> | 0.3 | 179.2k | 5.186 |
> | 0.4 | 153.6k | 4.676 |
> | 0.5 | 128k | 4.345 |
> | 0.6 | 102.4k | 3.722 |
> | 0.7 | 76.8k | 3.308 |
> | 0.8 | 51.2k | 3.071 |
> | 0.9 | 25.6k | 2.635 |
> | 1.0 | 0 | 0.955 |

---

> ### Author Response · Authors · 2025-11-20
> **Response to Reviewer 4HBM Weakness #4**
>
> ### [Reviewer 4HBM-W4] :
>
> > *The reported results measure only throughput, latency, and load variance, **lacking evaluation** of model quality (perplexity, accuracy, downstream metrics). So it is unclear whether the offline routing preserves semantic fidelity or harms inference correctness.*
> >
>
> ### [Response to 4HBM-W4] Our work is a system-level optimization that rigorously preserves the model integrity and mathematical equivalence.
>
> Rather than modifying the model architecture, weights, or token generation logic, our work Sem-MoE operates beneath the surface; it only orchestrates model and data device placement, while never altering the model’s internal computations or introducing approximations like quantization. Thus, an LLM’s output remains completely faithful to the original.

---

> ### Author Response · Authors · 2025-11-20
> **Response to Reviewer 4HBM Weakness #5**
>
> ### [Reviewer 4HBM-W5] :
>
> > All experiments are performed on a single 8×A100 NVLink node (§4). The paper suggests that the method could achieve larger gains under slower interconnects (PCIe or InfiniBand), but this claim remains untested. Including cross-node or heterogeneous-network experiments would strengthen the scalability argument.
> >
>
> ### [Response to 4HBM-W5]: Sem-MoE offers larger performance gains in interconnect-bandwidth-constrained settings.
>
> Unfortunately, due to hardware availability constraints, we do not have access to physical platforms with PCI-E interconnects or multi-node. To rigorously address this concern and evaluate how Sem-MoE’s performance scales with interconnect bandwidth, we conduct a simulation study using **three simulated hardware configurations:**
>
> - an 8×H100 system with **homogeneous, fast** NVLink interconnects **(900 GB/s)**
> - an 8×H100 system with **homogenous, slow** PCI-E interconnects **(128 GB/s)**
> - a dual-node system with **heterogeneous interconnects,** each node with 4 H100 GPUs interconnected **via NVLink (900 GB/s) ,** and the two nodes **interconnected by InfiniBand (limited to only 50GB/s)**
>
> They share identical computational capabilities and memory bandwidths, isolating interconnect speed as the primary variable.  Our simulation replays requests from real datasets and explicitly traces each token-expert routing event to capture the volume of remote communications. These routing events are then fed into an analytical cost model that combines abstracted computation cost with explicit communication overhead to compute end-to-end latency, enabling a quantitative comparison across interconnect regimes.
>
> Our simulation results for DeepSeek-Ve-Lite and Qwen3-30B on ShareGPT and LMSYS-CHAT-1M datasets (summarized in the tables below) demonstrate that **SemMoE's performance improvements are amplified in bandwidth-constrained environments**. On the high-speed NVLink platform, SemMoE reduces latency by an average of 17.38% across models and datasets. However, on the PCI-E platform with slower interconnects, the latency reduction increases to an average of 26%. Similarly, the latency reduction further increases to an average of 26.9% for the heterogeneous NVLink-IB setup. This confirms our approach effectively mitigates communication overheads and yields larger gains on commodity hardware with slower interconnects. **Overall, our approach is especially valuable for practitioners who cannot afford expensive high-bandwidth systems, enabling them to get more out of cost-effective infrastructure.**
>
> Table 6: Simulation: Evaluating Sem-MoE under different interconnects, DeepSeek-V2-Lite on ShareGPT
>
> |  | Fast Interconnect (NVLink: 900GB/s) | Slow Interconnect (PCI-E: 128 GB/s) | Dual-node Heterogeneous Interconnect (NVLink:900GB/s, IB:50GB/s) |
> | --- | --- | --- | --- |
> | SemMoE: Averaged E2E Latency (s) | 1.284 | 4.795 | 8.085 |
> | Baseline: Averaged E2E Latency (s) | 1.525 | 6.491 | 11.102 |
> | Latency Reduction | **15.8%** | **26.1%** | **27.2%** |
>
> &nbsp;
> &nbsp;
> &nbsp;
> &nbsp;
> &nbsp;
> &nbsp;
>
> Table 7: Simulation: Evaluating Sem-MoE under different interconnects, DeepSeek-V2-Lite on LMSYS-CHAT-1M
>
> |  | Fast Interconnect (NVLink: 900GB/s) | Slow Interconnect (PCI-E: 128 GB/s) | Dual-node Heterogeneous Interconnect (NVLink:900GB/s, IB:50GB/s) |
> | --- | --- | --- | --- |
> | SemMoE: Averaged E2E Latency (s) | 0.653 | 2.426 | 4.088 |
> | Baseline: Averaged E2E Latency (s) | 0.781 | 3.321 | 5.680 |
> | Latency Reduction | **16.3%** | **26.9%** | **28.0%** |
>
> &nbsp;
> &nbsp;
> &nbsp;
> &nbsp;
> &nbsp;
> &nbsp;
>
> Table 8: Simulation: Evaluating Sem-MoE under different interconnects, DeepSeek-V2-Lite on ShareGPT
>
> |  | Fast Interconnect (NVLink: 900GB/s) | Slow Interconnect (PCI-E: 128 GB/s) | Dual-node Heterogeneous Interconnect (NVLink:900GB/s, IB:50GB/s) |
> | --- | --- | --- | --- |
> | SemMoE: Averaged E2E Latency (s) | 2.445 | 11.361 | 19.602 |
> | Baseline: Averaged E2E Latency (s) | 3.064 | 15.711 | 27.346 |
> | Latency Reduction | **20.2%** | **27.7%** | **28.3%** |
>
> &nbsp;
> &nbsp;
> &nbsp;
> &nbsp;
> &nbsp;
> &nbsp;
>
> Table 9: Simulation: Evaluating Sem-MoE under different interconnect speeds, DeepSeek-V2-Lite on ShareGPT
>
> |  | Fast Interconnect (NVLink: 900GB/s) | Slow Interconnect (PCI-E: 128 GB/s) | Dual-node Heterogeneous Interconnect (NVLink:900GB/s, IB:50GB/s) |
> | --- | --- | --- | --- |
> | SemMoE: Averaged E2E Latency (s) | 1.221 | 5.781 | 9.992 |
> | Baseline: Averaged E2E Latency (s) | 1.475 | 7.562 | 13.162 |
> | Latency Reduction | **17.2%** | **23.5%** | **24.1%** |

---

> ### Author Response · Authors · 2025-11-20
> **Response to Reviewer 4HBM Weakness #6**
>
> ### [Reviewer 4HBM-W6] :
>
> > Consequently, the statistical input to the ILP may not reflect real routing behavior when **context distributions shift**.
> >
>
> > The paper lacks ablation studies testing ILP sensitivity to noise or **sampling bias in `C_{p,jk}`**
> >
>
> ### [Response to 4HBM-W6]: The ILP (scheduler) in Sem-MoE shows robust cross-dataset, out-of-distribution performance.
>
> Specifically, we trained Sem-MoE’s model (i.e., estimated  **`C_{p,jk}`** ) on one dataset and evaluated the resulting ILP solution on other unseen datasets that encompass unknown semantic and syntactic patterns, as well as different contextual distributions. The results demonstrate that our ILP-based optimization remains effective, achieving robust performance with only a minor degradation compared to in-distribution solutions.
>
> Specifically, we train Sem-MoE’s predictor on one dataset (e.g., ShareGPT) and evaluate our ILP-solution— Local Activation Rate (LAR)—on unseen datasets (e.g., Lmsys-Chat-1M and MMLU), for the two models DeepSeek-V2-Lite and Qwen3-30B.  Results show that while in-distribution ILP-solutions yield the highest locality (e.g., 46.49% for ShareGPT to ShareGPT), our out-of-distribution ILP-solution remains robust. For instance, our predictor trained from ShareGPT achieves a cross-dataset LAR 41.25% onto Lmsys-Chat-1M, which is **only a minor degradation compared to the in-distribution solution on Lmsys-Chat-1M (LAR: 47.19%)—** yet **it still significantly outperforms the SGLang defaults(**LAR: 25%). The same trend holds for all transfer directions and for Qwen3.
>
> In summary, the consistent performance across these out-of-distribution evaluations demonstrates that our ILP formulation is robust to variations in the underlying context distribution and does not overfit to the idiosyncrasies of a single data set.

---

> ### Author Response · Authors · 2025-11-20
> **Response to Reviewer 4HBM Questions**
>
> ### [Reviewer 4HBM-Q1] :
>
> > The paper defines “context-independent routing” (§3.2) based on the observed stability of token–expert activation frequencies computed from full-sentence attention representations. However, the gating function $$G_L(h_{L,j})$$ inherently depends on contextual hidden states. How do the authors justify that the offline probability $$P(E_k|x_j)$$ estimated under contextualized inputs approximates the true context-independent term $$P(E_k|x_j,h_{L,j})$$?
> >
>
> > In particular, what does “semantic independence” mean in this formulation—is it statistical stability of frequent tokens, or an actual invariance of routing under different semantic contexts?
> >
>
> ### [Response to 4HBM-Q1]:
>
> This question corresponds to the 1st point of weakness, i.e., **[Reviewer 4HBM-W1].** Please check our response above in **[Response to 4HBM-W1]**.
>
> ### [Reviewer 4HBM-Q2] :
>
> > The authors report stability (96.3%) and F1 (78.8%) on ShareGPT, but the analysis seems limited to a single layer and dataset. Could they clarify whether similar stability holds across different layers, seeds, or corpora?
> >
>
> ### [Response to 4HBM-Q2]:
>
> This question corresponds to the 1st point of weakness, i.e., **[Reviewer 4HBM-W1].** Please check our response above in **[Response to 4HBM-W1]**.
>
> ### [Reviewer 4HBM-Q3] :
>
> > Figures 6(b)(c) analyze only common function tokens. The text briefly mentions that rare tokens are “more uniformly distributed,” but no quantitative results are given. Could the authors elaborate on the behavior of low-frequency or semantic tokens?
> >
>
> ### [Response to 4HBM-Q3]:
>
> This question corresponds to the 2nd point of weakness, i.e., **[Reviewer 4HBM-W2].** Please check our response above in **[Response to 4HBM-W2]**.
>
> ### [Reviewer 4HBM-Q4] :
>
> > The author mentioned “the benefit will be even larger under slower interconnects (PCIe or InfiniBand)” in §4 and suggests that greater improvements would be observed under slower interconnects (PCIe or InfiniBand), yet this claim is untested. Have the authors evaluated or simulated multi-node scenarios to confirm that the ILP-based grouping remains effective when communication latency and bandwidth vary?
> >
>
> ### [Response to 4HBM-Q4]:
>
> This question corresponds to the 5th point of weakness, i.e., **[Reviewer 4HBM-W5].** Please check our response above in **[Response to 4HBM-W5]**.

---

> ### Author Response · Authors · 2025-11-27
> **Gentle Reminder**
>
> We sincerely thank you for your constructive feedback. As the discussion period is coming to a close, we would like to respectfully follow up to ensure that our responses and newly added experiments have fully addressed your concerns. We remain available to answer any further questions immediately.
>
> If our revisions meet your expectations, we kindly request that you consider reassessing your rating. Your evaluation plays a crucial role in the visibility and impact of our work. Thank you again for your time and thoughtful suggestions.

---

> ### Comment · Reviewer_4HBM · 2025-11-28
>
> Thank you for the detailed rebuttal. I have read your responses to several weaknesses, as well as the additional experimental results (including the context diversity of the hottest / top-70% frequency tokens, routing statistics for rare tokens, layer-wise F1 and hotness summaries on DeepSeek-V2-Lite, and the relationship between local activation rate and all-to-all latency), and reconsidered the paper in light of both the original submission and the new material.
>
> Taking the original experiments together with the supplementary ones, the evidence for routing stability is now clearly stronger than in the initial version. The original paper already visualized stable expert-usage patterns for some high-frequency tokens, and the new experiments systematically broaden this picture across token types, layers, and workloads, while also providing additional support that exploiting this stability leads to tangible system-level benefits.These results are consistent with a broader body of work that has observed relatively stable contextual embeddings and early functional specialization of MoE experts. In that sense, the original experiments plus the new ones can reasonably be read as a more detailed, data-specific quantification of the general idea that token representations and routing exhibit a certain degree of stability in practice.
>
> On this basis, if one adopts a straightforward systems/algorithms perspective, the method can be summarized as follows: the paper empirically observes, on real workloads (as documented in the submission and rebuttal), that token–expert activations exhibit stable statistical patterns; these patterns are then treated as a prior for routing and partitioning, used to construct $C_p​$ and the corresponding ILP, and subsequently applied in DP/TP settings to drive scheduling and communication optimization. Under this logic—“data-driven stable prior + engineering optimization”—the method is sound at that level: the prior is observable and stable for the models and datasets considered; this aligns with existing understanding of token representations and MoE behavior; and the original + rebuttal results together indicate that exploiting this prior yields concrete, measurable system benefits.
>
> My main reservations now are about the framing and assumption level of the claimed contribution, rather than the engineering logic itself. The paper (and rebuttal) often uses phrases such as “context-independent routing” and “semantic independence,” which naturally read as asserting a semantic-level claim that expert selection is essentially independent of context and primarily determined by token identity. This is a stronger statement than saying that, under a given data distribution, $P(\text{expert} \mid \text{token}, \text{context})​$ exhibits a stable statistical pattern.
>
> In my reading, the experiments clearly support the latter, workload-specific statement: for the chat workloads considered, $P(\text{expert} \mid \text{token}, \text{context})​$ is statistically stable and highly predictable, so that token–expert patterns form a useful prior that can be exploited. The text, however, often adopts language closer to a more general, semantic-level claim, without clearly delimiting which level of property is actually being assumed or demonstrated. This creates a mismatch between the level of the evidence (a stable, workload-dependent routing prior) and the level of some of the phrasing (which suggests a more context-invariant mechanism).
>
> From my perspective, this is not an issue that can be resolved simply by adding more experiments of the same kind; it is primarily about aligning the strength of the claims with the scope of the evidence. The current results are sufficient to support the claim that, for the workloads under consideration, there exists a stable and useful routing prior that can be exploited for system optimization, but they are not, in their present form, enough to substantiate stronger “semantic/context independence” type claims.
>
> Consequently, I interpret the contribution as being on solid ground when$C_p$is framed as a workload-specific empirical prior driving a systems optimization story, whereas a stronger, more general “semantic-aware, context-independent routing” mechanism claim would require a higher level of abstraction and additional support than is currently provided. In this sense, at the level of “algorithms / systems + stable prior,” the original paper plus the rebuttal provide a reasonably complete justification and the work has value, but the concerns tied to the stronger “semantic stability / context independence” framing remain unresolved for me, so I retain my original score and recommendation.

---

### Official Review · Reviewer_Rf5L · 2025-10-31

**Soundness:** 3
**Presentation:** 3
**Contribution:** 3
**Rating:** 6
**Confidence:** 2

**Summary:**

The paper proposes Sem-MoE, a semantic-aware model–data collaborative scheduling framework that improves inference efficiency for large-scale MoE models. It jointly optimizes expert placement (offline model scheduling) and token/request routing (online data scheduling) to reduce all-to-all communication in expert parallelism.

**Strengths:**

1. This paper addresses a key bottleneck in MoE inference—the high communication overhead associated with expert parallelism.
2. It introduces a novel idea that aligns expert placement with token–expert affinities through semantic-aware co-scheduling.
3. The proposed framework demonstrates a comprehensive design, effectively supporting both DP and TP configurations.

**Weaknesses:**

1. The experimental setup is relatively limited, with evaluations conducted only on DeepSeek and Qwen; I recommend validating the conclusions across a broader range of models and datasets.

2. The reported performance gains in E2E latency appear limited, which limits the practical significance of the proposed scheme.

**Questions:**

1. Beyond SGLang and MoETuner, were any additional baselines included for comparison? The current comparison seems limited.
2. Do tasks or queries from different knowledge domains require extra training to maintain accurate token–expert mappings?
3. How often must Algorithm 1 be executed in practice, and does this introduce notable or impractical computational overhead?
4. The paper mentions online inter-request data scheduling to proactively rebatch incoming requests. In the experiments, what stochastic process assumptions or models are used to generate the query arrivals?

---

> ### Author Response · Authors · 2025-11-20
> **Response to Reviewer Rf5L**
>
> We express our sincere gratitude to Reviewer Rf5L for the insightful review. We are glad that you found our collaborative scheduling framework to be a novel and effective solution for the critical communication bottleneck in MoE inference. We are also heartened by your recognition of our comprehensive system design that effectively supports both DP and TP configurations. We have carefully considered your suggestions to broaden the experimental scope and clarify practical significance, and we provide detailed responses to your questions below.
>
> *In addition, we kindly request you to consider raising the score if our clarifications have adequately addressed your concerns.*

---

> ### Author Response · Authors · 2025-11-20
> **Response to Reviewer Rf5L Weakness #1**
>
> ### [Reviewer Rf5L-W1] :
>
> > The experimental setup is relatively limited, with evaluations conducted only on DeepSeek and Qwen;  I recommend validating the conclusions across a broader range of models and datasets.
> >
>
> ### [Response to Rf5L-W1]:
>
> We agree that broader evaluation strengthens claims of generalizability, and we’d like to clarify why our current experiments provide evidence for Sem-MoE’s robustness and applicability.
>
> **Model and Dataset Coverage.**
>
> - We selected DeepSeek-V2-Lite and Qwen3-30B because they represent the two dominant MoE design paradigms used in practice today—both employ many fine-grained experts with sparse activation. This architecture is increasingly common across modern large language models (e.g., OpenAI’s GPT-OSS, Meta’ Llama 4 Maverick, Moonshot’s Kimi K2 and Tencent’s Hunyuan-Large), suggesting our findings are broadly relevant.
> - Our evaluation spans three distinct datasets: MMLU (knowledge-intensive questions), ShareGPT (real user chat logs), and Lmsys-Chat-1M (diverse open-domain prompts). Together, they cover a wide range of prompt lengths, semantic complexity, and user intents. Sem-MoE consistently improves inference performance across all three, confirming its effectiveness under varied workloads.
>
> **Robustness Across Datasets.** To further test generalization, we trained Sem-MoE’s predictor on one dataset and applied the resulting scheduling policy to *unseen* datasets with different styles, topics, and prompt structures. Remarkably, performance remained strong—even without retraining and fine-tuning. This demonstrates that our method captures fundamental patterns in how language prompts interact with MoE architectures, rather than memorizing quirks of a single data source.
>
> **Practical Constraints and Next Steps.** While we acknowledge the value of evaluating additional models and datasets, given the tight rebuttal timeline,  we leave expanding evaluations on additional models (e.g., GPT-OSS) and datasets (e.g., LongBench, WildChat) most likely to future work. If sufficient and sustained accelerator resources become available during the rebuttal period, we may post updated results for an additional model.
>
> In summary, our evaluation combines architecturally representative models, diverse real-world workloads. The consistent out-of-distribution performance further confirms that Sem-MoE’s benefits extend well beyond the specific settings tested.

---

> ### Author Response · Authors · 2025-11-20
> **Response to Reviewer Rf5L Weakness #2**
>
> ### [Reviewer Rf5L-W2] :
>
> > The reported performance gains in E2E latency appear limited, which limits the practical significance of the proposed scheme.
> >
>
> ### [Response to Rf5L-W2] : **Sem-MoE delivers tangible, not slim, performance gain**s
>
> Our MoE inference acceleration scheme, Sem-MoE, supports both Attention-DP+EP and Attention-TP+EP—two parallelization schemes that serve fundamentally different inference workloads. **Critically, Sem-MoE delivers tangible wins in both settings**: it dramatically boosts throughput for large-batch, high-throughput deployments (Attention-DP+EP) and substantially cuts prefill latency for small-batch, low-latency scenarios (Attention-TP+EP). These are not marginal improvements but targeted accelerations aligned with each configuration. Below, we detail and clarify the specific gains Sem-MoE achieves in each setting.
>
> For Athe Attention-DP+EP setting, Sem-MoE delivers a **2.78× throughput improvement** over the default SGLANG implementation. This gain stems from our co-design of expert routing and communication scheduling, which effectively alleviates inter-device bandwidth pressure that otherwise limits scalability in data-parallel attention combined with expert parallelism.
>
> For the Attention-TP+EP setting, Sem-MoE significantly accelerates the prefill phase, **reducing TTFT** by **28.89% on DeepSeek-V2-Lite** and **24.90% on Qwen3-30B**. The end-to-end latency gain appears more modest because decoding dominates total request time for long outputs. During decoding, each request generates only one token per step, resulting in a small communication volume; thus, communication is not the bottleneck, and our optimizations yield limited benefit in this phase. **Consequently, the strong prefill gains are diluted when averaged over full E2E latency**.
>
> This distinction matters less in practice than it might seem. With **prefill–decode (PD) disaggregation** now becoming the industry standard—where prefill and decode run on separate, specialized hardware pools—accelerating prefill alone is paramount. Therefore, even in attention-TP+EP settings, Sem-MoE provides substantial real-world value.
>
> In summary, **Sem-MoE delivers tangible, not slim, performance gains in both MoE inference deployments—just in different phases and forms aligned with their respective deployment objectives.

---

> ### Author Response · Authors · 2025-11-20
> **Response to Reviewer Rf5L Question #1**
>
> ### [Reviewer Rf5L-Q1] :
>
> > Beyond SGLang and MoETuner, were any additional baselines included for comparison? The current comparison seems limited.
> >
>
> ### [Response to Rf5L-Q1] :
>
> We fully acknowledge the value of comprehensive comparisons and appreciate the opportunity to clarify why the current selection of baselines—specifically **SGLang** and **MoETuner**—is **targeted and meaningful**. Our paper introduces a co-scheduling framework that jointly optimizes token dispatching and expert placement at inference time for MoE models, with the explicit goal of minimizing inter-device communication. To the best of our knowledge, there are currently limited existing systems that perform data-model co-scheduling for MoE serving, especially in a proactive joint-optimization manner.
>
> SGLang and MoETuner represent two **cutting-edge, state-of-the-art systems** for efficient MoE inference, each addressing a critical aspect of the data-model coordination problem. SGLang excels at **reactive, load-balancing–driven token scheduling** but assumes static expert placement and ignores cross-device communication or token-expert-routing predictability. MoETuner —also a recent and influential approach—takes a proactive stance by modeling empirical token–expert routing patterns to **optimize static expert placement,** thereby reducing communication, but does not co-optimize token dispatching at inference time. Together, they represent the current state of the art, each addressing only one half of the joint problem. Our work, Sem-MoE, unifies both dimensions through proactive, dynamic co-scheduling of tokens and experts, advancing beyond these important prior solutions.
>
> Besides SGLANG and MoETuner, we found that there exist other studies—accelerating MoE serving on memory-constrained hardware —that also leverage token–expert routing predictions —but primarily to decide which experts to pre-fetch from host memory. In memory-constrained settings where MoE experts are offloaded to CPU to reduce GPU memory usage, expert prefetching can yield modest acceleration by overlapping host-to-device transfers with computation, thereby mitigating—but not eliminating—latency overheads. While effective for reducing memory footprint, such methods inherently **incur significant latency penalties due to host-to-device data transfers and are not designed for—and often incompatible with—high-performance inference scenarios.** This line of work operates under fundamentally different assumptions and objectives than our work, which aims for low-latency, high-throughput MoE serving. For this reason, it is not directly comparable to our approach, and we do not include it as a baseline.

---

> ### Author Response · Authors · 2025-11-20
> **Response to Reviewer Rf5L Question #2**
>
> ### [Reviewer Rf5L-Q2] :
>
> > Do tasks or queries from different knowledge domains require extra training to maintain accurate token–expert mappings?
> >
>
> ### [Response to Rf5L-Q2] :Sem-MoE shows robust cross-dataset performance across distribution shifts.
>
> To assess the generalization of our method Sem-MoE, we conducted cross-domain experiments using two large language models: **DeepSeek-V2-Lite** and **Qwen3-30B**. Specifically, we trained Sem-MoE’s activation predictor on a single source dataset and evaluated the quality of its token-expert co-scheduling decisions, measured by the **Local Activation Rate (LAR)**, on the other two unseen, out-of-distribution target datasets. Note that LAR measures the proportion of tokens routed to local experts, reflecting a reduced communication volume to experts on remote devices. Therefore, a **higher** LAR indicates **lower** cross-device communication overhead—and is **better**.
>
> The results reveal **robust zero-shot transfer performance** for our method.
>
> - For **DeepSeek-V2-Lite**, the predictor trained on ShareGPT achieves an in-distribution LAR of **46.49%**. When applied without retraining or fine-tuning to Lmsys-Chat-1M, it still attains **41.25%** LAR—only a modest drop from the best in-domain result on that dataset (**47.19%**, achieved when training directly on Lmsys-Chat-1M). More importantly, this out-of-distribution performance is **~1.65× higher** than the SGLang’s default scheduling setting (**25.01%**). Similarly, on MMLU—a domain with very different content and structure—the same ShareGPT-trained predictor yields **41.55%** LAR, far surpassing SGLANG defaults (**24.97%**) and remaining close to in-distribution levels.
> - The trend is consistent for **Qwen3-30B**: the ShareGPT-trained predictor achieves **39.28%** LAR on Lmsys-Chat-1M and **35.30%** on MMLU, compared to peak in-domain scores of **43.61%** and **36.57%** (when trained on Lmsys), and both are substantially above the SGLang baseline (~25%).
> - A similar trend holds when transferring reservedly from Lmsys-Chat-1M to ShareGPT and MMLU.
>
> These results demonstrate that Sem-MoE may learn **generalizable routing patterns** that remain effective across different linguistic styles, task types, and knowledge domains. While training on the target distribution offers marginal improvements, it is **not required** except for extreme performance pursuits.
>
> Table 1: Cross-dataset evaluation of “zero-shot transfer” for DeepSeek-V2-Lite
>
> | Method | LAR on ShareGPT （p50) | LAR on Lmsys-Chat-1M (p50) | LAR on MMLU (p50) |
> | --- | --- | --- | --- |
> | Sem-MoE trained on ShareGPT | **46.49%** | 41.25% | 41.55% |
> | Sem-MoE trained on Lmsys-Chat-1m | 40.68% | **47.19%** | 41.40% |
> | Baseline: SGLang | 24.98% | 25.01% | 24.97% |
>
> Table 2:Cross-dataset evaluation of “zero-shot transfer” Qwen3-30B
>
> | Method | LAR on ShareGPT （p50) | LAR on Lmsys-Chat-1M (p50) | LAR on MMLU (p50) |
> | --- | --- | --- | --- |
> | Sem-MoE trained on ShareGPT | **46.88%** | 39.28% | 35.30% |
> | Sem-MoE trained on Lmsys-Chat-1m | 36.57% | **43.61%** | 33.37% |
> | Baseline: SGLang | 25.00% | 25.01% | 25.00% |

---

> ### Author Response · Authors · 2025-11-20
> **Response to Reviewer Rf5L Question #3**
>
> ### [Reviewer Rf5L-Q3] :
>
> > How often must Algorithm 1 be executed in practice, and does this introduce notable or impractical computational overhead?
> >
>
> ### [Response to Rf5L-Q3]:
>
>  **Computational cost**: Although we formulate the expert assignment problem as a 0-1 integer linear program (ILP) to establish a rigorous theoretical framework, we do **NOT rely on generic ILP solvers**—which are often impractical for large-scale or time-sensitive settings. Instead, we implement a **lightweight, greedy heuristic (described in Appendix B)** that efficiently approximates high-quality solutions with acceptable computational overhead.
>
> This design aligns well with real-world deployment constraints. For instance, our solver generates a new expert distribution plan in **<30 seconds** across different total numbers of experts (<256) and MoE layers (<100).  Crucially, this cost fits comfortably within existing system-level tolerances for dynamic MoE optimization. Modern inference engines like **SGLang** already provide native support for periodic dynamic MoE optimization. Following industry practice, systems typically trigger expert-redistribution (e.g., load re-balancing) every **1,000 iterations** [1]—a conservative interval chosen not only to amortize solver latency but also to mitigate performance interference from reloading expert weights or reshuffling routing tables.
>
> At an average iteration latency of **100 ms**, this yields a generous **1.67-minute (100-second) time window** between rebalancing events. Our solver’s runtime **fits the available budget**. Though **Sem-MoE** is algorithmically compatible with online periodic optimization—yet this is outside the scope of our current work due to practical reasons:
>
> - **Our approach shows fair generalization across workload shifts:** As aforementioned in [**Response to Rf5L-Q2**], Sem-MoE exhibits robust cross-dataset and out-of-distribution performance, as it tends to capture the invariability in token-routing patterns. Re-tuning the predictive model and expert distribution on a target workload is beneficial but can **NOT be an** **absolute necessity** except in scenarios where every last percentage point of efficiency is critical.
> - **Data collection overheads**: A fully online implementation requires a lightweight, high-throughput pipeline to collect, transfer, and aggregate per-token activation statistics from GPU to CPU with minimal latency and memory overheads. Designing such a monitoring subsystem is non-trivial and would substantially expand the system engineering effort.
>
> [1] https://docs.sglang.ai/advanced_features/server_arguments.html

---

> ### Author Response · Authors · 2025-11-20
> **Response to Reviewer Rf5L Question #4**
>
> ### [Reviewer Rf5L-Q4] :
>
> > The paper mentions online inter-request data scheduling to proactively rebatch incoming requests. In the experiments, what stochastic process assumptions or models are used to generate the query arrivals?
> >
>
> ### [Response to Rf5L-Q4]: **We employ a widely adopted, community-standard benchmarking tool to generate request arrivals.**
>
> **Evaluation Protocol:** To ensure fairness and reproducibility, we use the official `benchmark_serving` script [1] from the SGLang framework **without any modifications**. This script represents the de facto standard for evaluating modern large language model (LLM) serving systems and is extensively used in the community for performance comparisons.
>
> **Request Generation Mechanism:**
>
> - **Workload Distribution:** Prompts are selected from the test dataset via **random sampling with replacement**. This approach preserves the statistical characteristics of the original dataset while introducing natural variability in the input sequence.
> - **Arrival Pattern:** We emulate diverse load conditions by adjusting the **request arrival rate** and concurrency parameters within the benchmark script. This enables us to assess Sem-MoE’s performance across a wide range of traffic intensities—from low-load scenarios to system saturation.
>
> By adhering to the same stochastic assumptions as the official SGLang baseline—namely, random prompt selection and controlled arrival rates—we isolate the impact of our scheduling optimizations. Consequently, any observed performance improvements can be confidently attributed to our proposed enhancements rather than differences in workload generation methodology.
>
> [1] SGLang, "Benchmark Serving", https://github.com/sgl-project/sglang/pull/657

---

> ### Comment · Reviewer_Rf5L · 2025-11-27
> **Response to the authors**
>
> The authors have satisfactorily addressed my comments. I have no further comments and will maintain my rating.

---

> ### Author Response · Authors · 2025-11-27
> **Response to Reviewer Rf5L and Presentation of Additional Experiments on Moonlight-16B-A3B**
>
> Thanks for your comment. We are delighted to learn that our previous responses have satisfactorily addressed your concerns.
>
> Furthermore, to more thoroughly evaluate the generalization capabilities of our approach and fully address your suggestion regarding model diversity, we have successfully conducted additional experiments on the Moonlight-16B model. We respectfully present these added results below; it strengthen our belief that Sem-MoE can significantly enhance Mixture-of-Experts (MoE) inference performance and exhibits generality across models and workloads.
>
> **We kindly request that you consider updating your rating to reflect all our efforts. Your evaluation plays a crucial role in the visibility and impact of our work.**
>
>
>
> The added results for Moonlight-16B align closely with the performance trends for DeepSeek-V2-Lite and Qwen3-A30B, as reported in the manuscript. In summary, our approach Sem-MoE consistently outperforms SGLANG (MoETuner); it achieves an average performance improvement of 1.12× (1.22×) for PREFILL and 1.10× (1.14×) for E2E inference, respectively. Detailed results are provided in Table 10 and table 11.
>
> Table 10: Throughput Optimization Effect under TTFT SLO
> | Dataset | SGLang Throughput (Req/s) | MoETuner Throughput (Req/s) | Sem-MoE Throughput (Req/s) | Speedup v.s. SGLang | Speedup v.s. MoETuner |
> |----|----|----|----|----|----|
> |Lmsys-Chat-1M|32.7|28.6|**38.6**|1.18x|1.35x|
> |MMLU|32.2|30.1|**38.6**|1.20x|1.28x|
> |ShareGPT|2.9|2.8|**2.9**|1.00x|1.02x|
> | Average |22.6|20.5|**26.7**|1.12x|1.22x|
>
> Table 11: Throughput Optimization Effect under E2E latency SLO
> | Dataset | SGLang Throughput (Req/s) | MoETuner Throughput (Req/s) | Sem-MoE Throughput (Req/s) | Speedup v.s. SGLang | Speedup v.s. MoETuner |
> |----|----|----|----|----|----|
> |Lmsys-Chat-1M|37.2|35.5|**38.6**|1.04x|1.09x|
> |MMLU|37.9|35.3|**38.6**|1.02x|1.10x|
> |ShareGPT|1.9|1.9|**2.4**|1.23x|1.23x|
> | Average |25.7|24.2|**26.5**|1.10x|1.14x|

---

### Official Review · Reviewer_Nc2B · 2025-11-01

**Soundness:** 3
**Presentation:** 3
**Contribution:** 3
**Rating:** 6
**Confidence:** 5

**Summary:**

The Sem-MoE paper introduces an efficient method to enhance the performance of Mixture-of-Experts (MoE) models by reducing heavy communication between GPUs during inference. The authors do not treat model placement and data routing separately. Instead, they design a semantic-aware scheduler. This scheduler predicts which experts each token will likely activate. It aims to keep those experts and tokens on the same GPU. The method combines offline expert clustering with online request routing. It also uses customized communication kernels to minimize overhead. Experiments on models like DeepSeek-V2-Lite and Qwen3-30B-A3B show significant gains in throughput and latency, demonstrating the effectiveness of the approach. However, the method relies on static token–expert correlations, requires costly optimization, and may struggle to adapt to real-world, changing workloads. Overall, it is a smart and well-engineered step toward more efficient MoE inference.

**Strengths:**

1. The semantic-guided approach is intuitive and elegant. It leverages predictable token–expert correlations that were mostly ignored by other MoE schedulers.
2. Extensive experiments on DeepSeek-V2-Lite and Qwen3-30B-A3B show consistent and significant improvements in both throughput and latency.

**Weaknesses:**

1. The semantic prediction is based on historical activation statistics. In real-world, open-domain inference, input distributions can shift significantly. Such shifts may invalidate prior correlations and reduce performance.
2. The optimization process is largely offline and static. When the model or workload changes, Sem-MoE requires re-optimization, which can be computationally expensive.
3. Although the method is non-intrusive, Sem-MoE modifies the runtime scheduler and introduces custom collective operations. This makes integration into production-level inference frameworks more complex than suggested.
4. There is no analysis of energy consumption or communication bandwidth efficiency in the paper.
6.  The author should consider more types of MoE LLMs, such as Qwen3, Moonlight-A3B, gpt-oss-120b, and gpt-oss-20b, should be considered.

**Questions:**

Summarized in Weakness.

---

> ### Author Response · Authors · 2025-11-20
> **Response to Reviewer Nc2B**
>
> We sincerely thank Reviewer Nc2B for your thoughtful and constructive feedback. We are highly encouraged by your appreciation of our semantic-guided approach as "intuitive and elegant" and your recognition that utilizing predictable token-expert correlations effectively addresses gaps left by prior schedulers. We also value your positive note on the consistent and significant improvements demonstrated in our experiments. We appreciate your suggestions for further strengthening the paper and addressing each of your concerns in detail below.
>
> *In addition, we kindly request you to consider raising the score if our clarifications have adequately addressed your concerns.*

---

> ### Author Response · Authors · 2025-11-20
> **Response to Reviewer Nc2B Weakness #1**
>
> ### [Reviewer Nc2B-W1] :
>
> > The semantic prediction is based on historical activation statistics. In real-world, open-domain inference, input distributions can shift significantly. Such shifts may invalidate prior correlations and reduce performance.
> >
>
> ### [Response to Nc2B-W1] Sem-MoE shows robust cross-dataset, out-of-distribution performance across distribution shifts.
>
> To assess the generalization of our method Sem-MoE, we conducted cross-domain experiments using two large language models: **DeepSeek-V2-Lite** and **Qwen3-30B**. Specifically, we trained Sem-MoE’s activation predictor on a single source dataset and evaluated the quality of its token-expert co-scheduling decisions, measured by the **Local Activation Rate (LAR)**, on the other two unseen, out-of-distribution target datasets. Note that LAR measures the proportion of tokens routed to local experts, reflecting a reduced communication volume to experts on remote devices. Therefore, a **higher** LAR indicates **lower** cross-device communication overhead—and is **better**.
>
> The results reveal **robust zero-shot transfer performance** for our method.
>
> - For **DeepSeek-V2-Lite**, the predictor trained on ShareGPT achieves an in-distribution LAR of **46.49%**. When applied without retraining or fine-tuning to Lmsys-Chat-1M, it still attains **41.25%** LAR—only a modest drop from the best in-domain result on that dataset (**47.19%**, achieved when training directly on Lmsys-Chat-1M). More importantly, this out-of-distribution performance is **~1.65× higher** than the SGLang’s default scheduling setting (**25.01%**). Similarly, on MMLU—a domain with very different content and structure—the same ShareGPT-trained predictor yields **41.55%** LAR, far surpassing SGLANG defaults (**24.97%**) and remaining close to in-distribution levels.
> - The trend is consistent for **Qwen3-30B**: the ShareGPT-trained predictor achieves **39.28%** LAR on Lmsys-Chat-1M and **35.30%** on MMLU, compared to peak in-domain scores of **43.61%** and **36.57%** (when trained on Lmsys), and both are substantially above the SGLang baseline (~25%).
> - A similar trend holds when transferring reservedly from Lmsys-Chat-1M to ShareGPT and MMLU.
>
> These results demonstrate that when fed with real-world, representative datasets, Sem-MoE tends to capture the **invariability in token-routing patterns** that remain effective across different linguistic styles, task types, and knowledge domains. While training on the target distribution offers marginal improvements, it is **not required** except for extreme performance pursuits.
>
> Table 1: Cross-dataset evaluation of “zero-shot transfer” for DeepSeek-V2-Lite
>
> | Method | LAR on ShareGPT （p50) | LAR on Lmsys-Chat-1M (p50) | LAR on MMLU (p50) |
> | --- | --- | --- | --- |
> | Sem-MoE trained on ShareGPT | **46.49%** | 41.25% | 41.55% |
> | Sem-MoE trained on Lmsys-Chat-1m | 40.68% | **47.19%** | 41.40% |
> | Baseline: SGLang | 24.98% | 25.01% | 24.97% |
>
> &nbsp;
>
> Table 2:Cross-dataset evaluation of “zero-shot transfer” Qwen3-30B
>
> | Method | LAR on ShareGPT （p50) | LAR on Lmsys-Chat-1M (p50) | LAR on MMLU (p50) |
> | --- | --- | --- | --- |
> | Sem-MoE trained on ShareGPT | **46.88%** | 39.28% | 35.30% |
> | Sem-MoE trained on Lmsys-Chat-1m | 36.57% | **43.61%** | 33.37% |
> | Baseline: SGLang | 25.00% | 25.01% | 25.00% |

---

> ### Author Response · Authors · 2025-11-20
> **Response to Reviewer Nc2B Weakness #2**
>
> ### [Reviewer Nc2B-W2] :
>
> > The optimization process is largely offline and static. When the model or workload changes, Sem-MoE requires re-optimization, which can be computationally expensive.
> >
>
> ### [Response to Nc2B-W2] **Sem-MoE is algorithmically compatible with online periodic optimization—yet it is** outside the scope of our current work due to practical reasons.
>
> We clarify that **Sem-MoE is algorithmically compatible with online periodic optimization.** Our current use of offline scheduling stems from practical considerations—not algorithmic limitations.
>
> **Online integration for dynamic re-optimization is feasible.** SGLAGNG  already provides the infrastructure needed to support periodic redistribution of MoE experts over accelerator devices, making online deployment of Sem-MoE feasible.
>
> - **Native support in SGLANG**: Open-source engines like **SGLang** natively support dynamic expert redistribution for load rebalancing purposes, with redistribution frequency controlled by `--eplb-rebalance-num-iterations`.
> - **Ample time budget**: Industry practice typically sets rebalancing intervals at **1,000 iterations** [1] to tolerate solver overheads and the interference induced by reloading expert weights. At an average iteration latency of 100 ms, this yields a **1.67min time budget** to re-solve an expert distribution plan.
> - **Acceptable optimization overhead of our solver**: Our solver (See our fast heuristic algorithm in Appendix B) generates a new expert placement plan in **<30 seconds** across different total numbers of experts (<256) and MoE layers (<100). This demonstrates that Sem-MoE can perform online re-optimization **within the allowed time budget in existing inference engines**.
>
> Our practical considerations for employing an offline schedule:
>
> - **Our approach shows fair generalization across workload shifts:** As aforementioned in [**Response to Nc2B-W1**], Sem-MoE exhibits robust cross-dataset and out-of-distribution performance, as it tends to capture the invariability in token-routing patterns. Re-tuning the predictive model and expert distribution on a target workload yields gains but is **NOT an** **absolute necessity** except in scenarios where every last percentage point of efficiency is critical.
> - **Data collection overheads**: A fully online implementation requires a lightweight, high-throughput pipeline to collect, transfer, and aggregate per-token activation statistics from GPU to CPU with minimal latency and memory overheads. Designing such a monitoring subsystem is non-trivial and would substantially expand the system engineering effort.
>
> We view the development of a pure online version of Sem-MoE as valuable **future work** that will unlock Sem-MoE’s full dynamic potential.
>
> [1] https://docs.sglang.ai/advanced_features/server_arguments.html

---

> ### Author Response · Authors · 2025-11-20
> **Response to Reviewer Nc2B Weakness #3**
>
> ### [Reviewer Nc2B-W3] :
>
> > Although the method is non-intrusive, Sem-MoE modifies the runtime scheduler and introduces custom collective operations. This makes integration into production-level inference frameworks more complex than suggested.
> >
>
> ### [Response to **Nc2B**-W3] Sem-MoE’s **minor, non-intrusive changes to existing frameworks align with production deployment needs.**
>
> For the **DP-Attention** setting, Sem-MoE preserves the original scheduling logic of the underlying SGLANG framework, requiring minimal code changes to add our DP scheduler as a pluggable policy module into SGLANG’s DP controller [1]. To enable our scheduler,  we just need to simply update an argument `--scheduling-policy` that specifies the scheduling policy when starting an instance of the SGLANG.  The expert re-distribution functionality is already supported by SGLANG [2][3], and we directly use this feature without changing any source code. These engineering considerations make Sem-MoE **convenient** to deploy and **smoothly** integrated in production environments without disrupting existing workflows of the SGLANG inference engine.
>
> [1] https://github.com/sgl-project/sglang/blob/main/python/sglang/srt/managers/data_parallel_controller.py
>
> [2] https://github.com/sgl-project/sglang/issues/5309
>
> [3] https://docs.sglang.ai/advanced_features/server_arguments.html
>
> For the **TP-Attention** setting, the “custom” collective operations are not new communication primitives but **thin, composable wrappers** around standard  `reduce-scatter` and `all-gather` calls—operations already heavily optimized and widely used in frameworks in SGLang. These wrappers only add a modest amount of metadata and their simple operations and do not alter the underlying data movement and communication logic. As a result, they incur **no measurable latency overhead** and preserve compatibility with existing tensor parallelism pipelines.

---

> ### Author Response · Authors · 2025-11-20
> **Response to Reviewer Nc2B Weakness #4**
>
> ### [Reviewer Nc2B-W4] :
>
> > There is no analysis of energy consumption or communication bandwidth efficiency in the paper.
> >
>
> ### [Response to Nc2B-W4]: Our primary optimization metrics serve as strong and well-established proxies for both energy and bandwidth efficiency.
>
> We agree that energy consumption and communication bandwidth efficiency are critical considerations for deploying large language models in real-world systems. While our paper does not include these direct measurements from hardware, we emphasize that our primary optimization metrics serve as strong and well-established proxies for both energy and bandwidth efficiency:
>
> 1. **Energy Efficiency:** Sem-MoE directly lowers dynamic power consumption; our reported **2.78× improvement in throughput** translates to a proportional reduction in GPU active time per token, which in turn reduces total energy consumption per generated token.
> 2. **Bandwidth Efficiency:** Inter-device communication is among the most energy-intensive operations in distributed inference. Our analysis explicitly quantifies the reduction in **all-to-all communication traffic**.  Sem-MoE reduces remote token routing over the network fabric by **37–43%**, minimizing unnecessary cross-device data transfers. This maximizes the utilization of limited interconnect bandwidth for only essential remote activations, thereby improving overall bandwidth efficiency.

---

> ### Author Response · Authors · 2025-11-20
> **Response to Reviewer Nc2B Weakness #5**
>
> ### [Reviewer Nc2B-W5]:
>
> > The author should consider more types of MoE LLMs, such as Qwen3, Moonlight-A3B, gpt-oss-120b, and gpt-oss-20b, should be considered.
> >
>
> ### [Response to Nc2B-W5]:
>
> We agree that broader evaluation strengthens claims of generalizability, and we’d like to clarify why our current experiments provide evidence for Sem-MoE’s robustness and applicability.
>
> **Model and Dataset Coverage.**
>
> - We selected DeepSeek-V2-Lite and Qwen3-30B because they represent the two dominant MoE design paradigms used in practice today—both employ many fine-grained experts with sparse activation. This architecture is increasingly common across modern large language models (e.g., OpenAI’s GPT-OSS, Meta’s Llama 4 Maverick, Moonshot’s Kimi K2 and Moonligh-A3B, and Tencent’s Hunyuan-Large), suggesting our findings are broadly relevant.
> - Our evaluation spans three distinct datasets: MMLU (knowledge-intensive questions), ShareGPT (real user chat logs), and Lmsys-Chat-1M (diverse open-domain prompts). Together, they cover a wide range of prompt lengths, semantic complexity, and user intents. Sem-MoE consistently improves inference performance across all three, confirming its effectiveness under varied workloads.
>
> **Robustness Across Datasets.** To further test generalization, we trained Sem-MoE’s predictor on one dataset and applied the resulting scheduling policy to *unseen* datasets with different styles, topics, and prompt structures. Remarkably, performance remained strong—even without retraining and fine-tuning. This demonstrates that our method captures fundamental patterns in how language prompts interact with MoE architectures, rather than memorizing quirks of a single data source.
>
> **Practical Constraints and Next Steps.** While we acknowledge the value of evaluating additional models and datasets, given the tight rebuttal timeline,  we leave expanding evaluations on additional models (e.g., GPT-OSS) and datasets (e.g., LongBench, WildChat) most likely to future work. If sufficient and sustained accelerator resources become available during the rebuttal period, we may post updated results for an additional model.
>
> In summary, our evaluation combines architecturally representative models, diverse real-world workloads. The consistent out-of-distribution performance further confirms that Sem-MoE’s benefits extend well beyond the specific settings tested.

---

> ### Author Response · Authors · 2025-11-27
> **Response to Reviewer Nc2B regarding Model Diversity (Weakness #5): Additional Experiments**
>
> [Reviewer Nc2B-W1] :
> > The author should consider more types of MoE LLMs, such as Qwen3, Moonlight-A3B, gpt-oss-120b, and gpt-oss-20b, should be considered.
>
> [Response to Reviewer Nc2B-W1] :
>
> To address the reviewer's concern regarding model diversity, we have successfully secured additional computing resources and, as suggested, incorporated benchmarking results for the Moonlight-16B model. The new results align closely with the performance trends observed for DeepSeek-V2-Lite and Qwen3-A30B, as reported in the manuscript. In summary, our approach Sem-MoE, consistently outperforms SGLANG (MoETuner): it achieves an average performance improvement of 1.12× (1.22×) for prefill and 1.10× (1.14×) for end-to-end inference, respectively. Detailed results are provided in Table 10 and table 11.
>
> Table 10: Throughput Optimization Effect under TTFT SLO
> | Dataset | SGLang Throughput (Req/s) | MoETuner Throughput (Req/s) | Sem-MoE Throughput (Req/s) | Speedup v.s. SGLang | Speedup v.s. MoETuner |
> |----|----|----|----|----|----|
> |Lmsys-Chat-1M|32.7|28.6|**38.6**|1.18x|1.35x|
> |MMLU|32.2|30.1|**38.6**|1.20x|1.28x|
> |ShareGPT|2.9|2.8|**2.9**|1.00x|1.02x|
> | Average |22.6|20.5|**26.7**|1.12x|1.22x|
>
> Table 11: Throughput Optimization Effect under E2E latency SLO
> | Dataset | SGLang Throughput (Req/s) | MoETuner Throughput (Req/s) | Sem-MoE Throughput (Req/s) | Speedup v.s. SGLang | Speedup v.s. MoETuner |
> |----|----|----|----|----|----|
> |Lmsys-Chat-1M|37.2|35.5|**38.6**|1.04x|1.09x|
> |MMLU|37.9|35.3|**38.6**|1.02x|1.10x|
> |ShareGPT|1.9|1.9|**2.4**|1.23x|1.23x|
> | Average |25.7|24.2|**26.5**|1.10x|1.14x|

---

> ### Author Response · Authors · 2025-11-27
> **Gentle Remainder**
>
> We thank the reviewer for the constructive feedback and would like to respectfully remind the reviewer that the deadline for the discussion period is approaching. If the reviewer has any other questions, we would be glad to address them.
>
> If the revisions meet your expectations, we kindly request you to reconsider raising your rating score.  Your evaluation plays a crucial role in the visibility and impact of our work.Thank you again for your time and thoughtful suggestions.

---

### Meta-Review · Area_Chair_iY9m · 2026-01-07

**Summary:**

The paper proposes Sem-MoE, a semantic-aware model–data collaborative scheduling framework that improves inference efficiency for large-scale MoE models. It jointly optimizes expert placement (offline model scheduling) and token/request routing (online data scheduling) to reduce all-to-all communication in expert parallelism. Experiments on models like DeepSeek-V2-Lite and Qwen3-30B-A3B show significant gains in throughput and latency, demonstrating the effectiveness of the approach. This paper makes a clear and meaningful contribution to the field, and the identified weaknesses do not undermine the validity or significance of the main results.

**Reviewer Concerns:**

Most concerns raised by reviewers are well-addressed by the author during rebuttal.

- The semantic prediction is based on historical activation statistics. This approach may face issues related to distribution shifts. The authors provide additional experiments, but including more datasets and model backbones in the final paper would be more helpful.

**Reviewer Scores:**

The authors have well addressed the reviewers’ questions during the rebuttal phase, and most reviewers are likely to improve their scores.

---

### Decision · Program_Chairs · 2026-01-26

Accept (Poster)